# AutoSafe: A Safe Neural Policy Network with Safety Common Sense

## Abstract

Recognizing and avoiding danger is a fundamental capability of biological intelligence, yet this principle is less explored in the design of neural policies in today's artificial intelligence. We present **AutoSafe**, a novel architecture that embeds *safety common sense* directly into neural policies for safety sensitive applications. In particular, **AutoSafe** integrates a lightweight model-based *Safety Evaluation Module* that continuously evaluates risk of safety violations and leverages a model-based safe policy as *Safety Correction Module* to correct potentially unsafe actions at runtime. By incorporating these two designs as part of the policy itself, **AutoSafe** can be seamlessly integrated into *actor-critic* based reinforcement learning algorithms to maximize the performance while maintaining safety. We evaluate **AutoSafe** on a suite of continuous-control benchmarks, demonstrating that **AutoSafe** consistently outperforms other safe reinforcement learning baselines. Finally, we showcase the applicability of the proposed architecture in a real-world continual learning scenario on a cartpole system.

## 1 Introduction

Standard deep reinforcement learning (DRL) policies map observations directly to actions when interacting with environments. However, such end-to-end inference provides limited safety assurance in safety sensitive applications (Kirk et al., 2023; Gu et al., 2024).

To ensure safe interactions, existing approaches either leverage an external safety layer (e.g., shields, filters, barrier-based controllers) to safeguard the system (Cheng et al., 2019a; Alshiekh et al., 2018; Zhong et al., 2023; Cao et al.; Cheng et al., 2019b) or incorporate safety requirements into the learning objectives (Westenbroek et al., 2022; Achiam et al., 2017; Zhao et al., 2023) that incentivize safe behavior. While effective in certain settings, external safety layer strategies often constrain the agent's performance, and safety-aware objectives typically lack formal safety guarantees. Overall, designing a safe policy without compromising performance is still an open challenge.

In this work, we draw inspiration from human safety behaviors, for example, a human driver who continuously monitors the environment and adaptively decides when and how strongly to brake when approaching an obstacle. We refer to this intuitive and risk-aware decision making as "safety common sense." Building on this idea, we extract such knowledge from a linear approximation of the system dynamics and propose a novel architecture, **AutoSafe**, that embeds this structured safety knowledge directly into the neural policy.

Specifically, **AutoSafe** integrates a lightweight, model-based *Safety Evaluation Module* grounded in the notion of a *Safety Envelope* (Sha, 2001). This module continuously evaluates risk of safety violations and, when necessary, leverages the *Safety Correction Module* (a model-based safe policy) to correct potentially unsafe actions. When the risk is high, **AutoSafe** relies entirely on the model-based safe policy for system control and protection. Conversely, when the risk is low, it prioritizes the learning-based policy to maximize performance.

Our architecture treats the safety constraint problem and the performance maximization problem separately as in *Safety Filter*Hsu et al. (2024). Safety constraints are handled explicitly through a model-based safety module, which contrasts with the Constrained Markov Decision Process (CMDP)-based frameworks (Altman, 2021; Achiam et al., 2017; Ha et al., 2020), where safety and performance are jointly optimized and often lead to degraded performance. Furthermore, the lightweight design of

**AutoSafe** bypasses the computational overhead of model-predictive control (MPC)–based approaches (Wabersich and Zeilinger, 2019; Tian et al., 2024), which demand online trajectory optimization, and offers greater flexibility than control barrier function (CBF)–based methods (Cheng et al., 2019c; Grandia et al., 2021), which require accurate affine models to construct safety certificates. Unlike related approaches that interface with a safety envelope or policy only externally (Cheng et al., 2019b;a; Alshiekh et al., 2018; Sha, 2001), **AutoSafe** incorporates both the safety envelope and the safety controller as internal components of the policy itself. This design enables **AutoSafe** to be seamlessly integrated with *actor-critic* based DRL algorithms and more efficiently trained to maximize the expected return.

Our main contributions are summarized as follows: 1) We propose **AutoSafe**, a novel risk-aware safe policy architecture equipped with integrated risk monitoring and safe action correction, enabling a smooth and continuous transition between safe and unsafe operational regimes; 2) We provide safety assurance with performance analysis, and we present extensive empirical validation and insights across a suite of simulated continuous control tasks; 3) We demonstrate the effectiveness of **AutoSafe** by deploying it on an embedded device for a real-world continual learning task on a cartpole system.

## 2 PRELIMINARIES

### 2.1 MARKOV DECISION PROCESS (MDP) AND SAFE DEEP REINFORCEMENT LEARNING

We formulate the agent-environment interaction as an infinite discount Markov Decision Process, defined as $\mathcal{M} = \{\mathcal{S}, \mathcal{A}, P, R, \gamma\}$, where $\mathcal{S} \subseteq \mathbb{R}^n$ is the $n$-dimensional state space, $\mathcal{A} \subseteq \mathbb{R}^m$ is the $m$-dimensional action space, $P : \mathcal{S} \times \mathcal{A} \to \mathcal{S}$ is the transition function that describes the system dynamics, $R : \mathcal{S} \times \mathcal{A} \to \mathbb{R}$ is the bounded reward function that maps state and action into a scalar value, $\gamma$ is the discount factor. The goal of safe DRL problem is to find a policy $\pi_\theta : \mathcal{S} \to \mathcal{A}$ that maximizes the expected return along a trajectory $\tau$, subject to the safety constraints in state space and action space at every step $t$, formally:

$$\text{maximize} : V(\mathbf{s}) = \mathbb{E}_{\tau \sim \pi_\theta}\left[\sum_{t=0}^{\infty} \gamma^t R(\mathbf{s}_t, \mathbf{a}_t)\Big|\mathbf{s}_0 = \mathbf{s}\right], \tag{1}$$

$$\text{subject to} : \mathbf{A}_s\mathbf{s}_t \leq \mathbf{b}_s, \ \mathbf{A}_a\mathbf{a}_t \leq \mathbf{b}_a, \ \forall t \in [0, \infty), \tag{2}$$

where $\tau = \{\mathbf{s}_0, \mathbf{a}_0, \mathbf{s}_1, \mathbf{a}_1, \ldots\}$ denotes the trajectory induced by policy $\pi_\theta$.

### 2.2 SAFETY SET, SAFE POLICY AND SAFETY ENVELOPE

The safety constraints mentioned in eq. (2) constitute a subset $\mathcal{S}_f \subset S$, known as the safety set. $\mathcal{S}_f$ is defined as $\mathcal{S}_f := \{\mathbf{s} \in \mathbb{R}^n | \mathbf{A}_s\mathbf{s} \leq \mathbf{b}_s\}$, with $\mathbf{A}_s \in \mathbb{R}^{n_s \times n}$, $\mathbf{b}_s \in \mathbb{R}^{n_s}$, where $\mathbf{A}_s$ and $\mathbf{b}_s$ represent $n_s$ polytopic constraints on the state space. Besides, the action space $\mathcal{A}$ is constrained by some actuation limits, from which the feasible action space $\mathcal{A}_f$ is defined as $\mathcal{A}_f := \{\mathbf{a} \in \mathbb{R}^m | \mathbf{A}_a\mathbf{a} \leq \mathbf{b}_a\}$, with $\mathbf{A}_a \in \mathbb{R}^{n_a \times m}$, $\mathbf{b}_a \in \mathbb{R}^{n_a}$, where $n_a$ is number of the action constraints. Given the defined safety set $\mathcal{S}_f$ and action set $\mathcal{A}_f$, we can search for a safe policy $\pi_{\text{safe}}$ that renders a subset $\mathcal{S}_c \subset \mathcal{S}_f$ forward invariant by considering partial **known** knowledge of the system dynamics in eq. (3):

$$\dot{\mathbf{s}} = \underbrace{\mathbf{As} + \mathbf{Ba}}_{known} + \underbrace{\mathbf{f}(\mathbf{s}, \mathbf{a})}_{unknown} \tag{3}$$

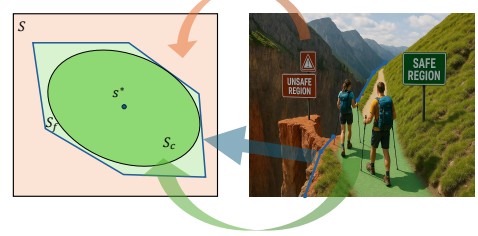

Figure 1: **Illustration of a static 2D case**: The state space $\mathcal{S}$ in the left picture corresponds to all movable region in the right; the safety set $\mathcal{S}_f$ (**polygon**) refers to the areas within the edges, and the safety envelope $\mathcal{S}_c \subset S_f$ (**ellipsoid**) means the inner safe region.

where $\dot{\mathbf{s}}$ denotes the time derivative of $\mathbf{s}$, $\mathbf{A} \in \mathbb{R}^{n \times n}$ is the nominal system matrix, $\mathbf{B} \in \mathbb{R}^{n \times m}$ control matrix, $\mathbf{a} \in \mathcal{A}_f$ is the applied control action, $\mathbf{f}(\mathbf{s}, \mathbf{a}) \in \mathbb{R}^n$ is the unknown model mismatch.

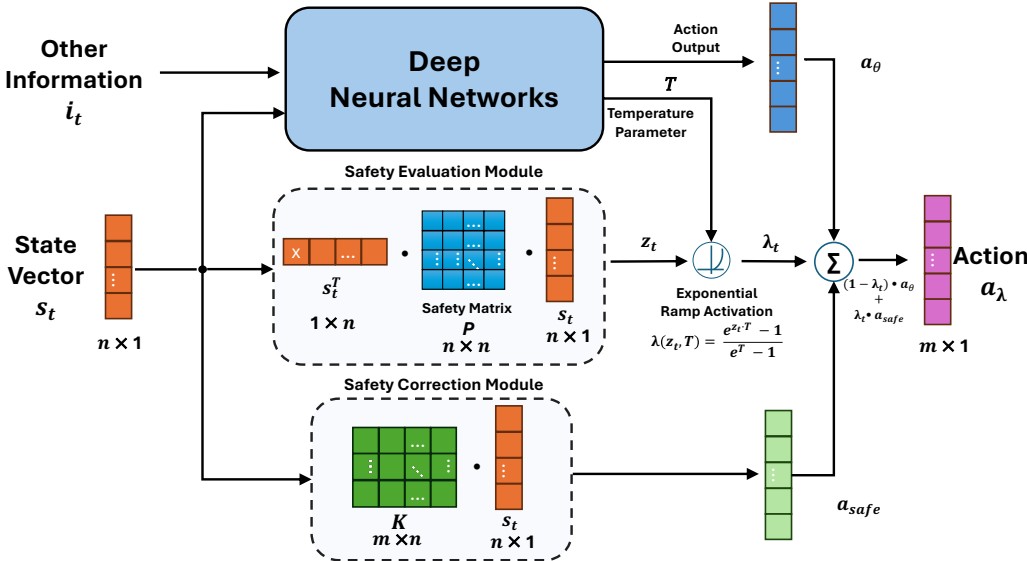

Figure 2: **AutoSafe**: The policy uses the state vector $\mathbf{s}_t$, included in the observation, to assess risk $z_t$ and generate safe action $\mathbf{a}_{\text{safe}}$. In parallel, the learning-based component processes the full observation $\mathbf{o}_t = (\mathbf{s}_t, \mathbf{i}_t)$ to produce high-performance but potentially unsafe actions $\mathbf{a}_\theta$. The two action outputs are fused through a weighted summation, where the weights are determined by a learnable exponential activation function conditioned on the risk value.

The set $\mathcal{S}_c$, called the safety envelope Sha (2001), is depicted as the ellipsoid in Figure 1, which is formally defined as below:

$$\text{Safety Envelope}: \ \mathcal{S}_\mathbf{c} \triangleq \left\{ \mathbf{s} \in \mathbb{R}^n \,|\, \mathbf{s}^\top \mathbf{P} \mathbf{s} \leq 1 \right\}, \tag{4}$$

where $\mathbf{P} \in \mathbb{R}^{n \times n}$ is a positive definite matrix that characterizes the shape of $\mathcal{S}_c$.

The obtained safe policy $\pi_{\text{safe}}$ is essentially a feedback control law $\mathbf{K} \in \mathbb{R}^{m \times n}$, which generates safe action as $\mathbf{a}_{\text{safe}} = \mathbf{K}\mathbf{s}$ (assume the equillibrium point $\mathbf{s}^*$ to be $\mathbf{0}$, without loss of generality). The solution for $\mathbf{P}$ and $\mathbf{K}$ can be found by using convex optimization tools, such as (Grant et al., 2009). We summarize the formulation of the optimization problem used in this work in section A.

## 3 AUTOSAFE

We introduce **AutoSafe**, a safe neural policy architecture that integrates a *Safety Evaluation Module*, based on the distance to the safety envelope $\mathcal{S}_c$, and a *Safety Correction Module*, which leverages the safe policy $\pi_{\text{safe}}$ to automatically generate safe actions in the presence of risk of safety violations.

### 3.1 ARCHITECTURE OVERVIEW

**AutoSafe** fuses a standard DRL policy $\pi_\theta$ with a safe model-based policy $\pi_{\text{safe}}$ through a weighted summation of their action outputs: $\mathbf{a}_\lambda = (1 - \lambda(\mathbf{s})) \cdot \mathbf{a}_\theta(\mathbf{o}) + \lambda(\mathbf{s}) \cdot \mathbf{a}_{\text{safe}}(\mathbf{s})$. The DRL policy $\pi_\theta$ takes the full observation $\mathbf{o}_t = (\mathbf{s}_t, \mathbf{i}_t)$, which includes the state vector $\mathbf{s}_t$ and additional information $\mathbf{i}_t$ (e.g., images), and generates an action $\mathbf{a}_\theta \sim \pi_\theta(\cdot \mid \mathbf{o}_t)$. The safe policy $\pi_{\text{safe}}$ only observes the state vector and outputs a safe action $\mathbf{a}_{\text{safe}} = \mathbf{K}\mathbf{s}$.

The mixing coefficient $\lambda(\mathbf{s}) \in [0, 1]$ is determined by an exponential-like activation function that takes as input a non-negative risk score $z \in \mathbb{R}_{\geq 0}$, computed by a non-learnable *Safety Evaluation Module*, together with a learnable temperature parameter $T$. When the agent operates well inside the safety envelope, $\lambda(\mathbf{s})$ is small and the action is dominated by $\mathbf{a}_\theta$. As the risk $z$ increases near the boundary of the safety set $\mathcal{S}_c$, $\lambda(\mathbf{s})$ grows, shifting the action towards $\mathbf{a}_{\text{safe}}$. At the boundary, the agent relies solely on the safe policy to ensure constraint satisfaction. The temperature parameter $T$

controls the sharpness of this transition. An overview of the entire architecture is shown in Figure 2, and the designs of individual components are described in the following sections.

## 3.2 SAFETY EVALUATION MODULE

Recognizing risk of safety violations during agent–environment interactions is a prerequisite for ensuring safe behavior. To this end, **AutoSafe** employs a Safety Evaluation Module (SEM) that quantifies risk based on the system state at each time step $t$. Specifically, SEM computes a risk score $z_t(\mathbf{s}_t) = \mathbf{s}_t^\top \mathbf{P} \mathbf{s}_t$, where $\mathbf{P} \succ 0$ is the safety matrix defined in section 2.2. The boundary of $\mathcal{S}_c$ is given by the condition $\mathbf{s}_t^\top \mathbf{P} \mathbf{s}_t = 1$. Intuitively, $z(\mathbf{s}_t) = 1$ indicates that the system state lies exactly on the boundary of $\mathcal{S}_c$. When $\mathbf{s}_t$ moves away from the ellipsoid center $\mathbf{s}^*$, the risk score $z(\mathbf{s}_t)$ increases, whereas actions that move the controlled system closer to $\mathbf{s}^*$ decrease the score. This risk score will be used to determine the contribution ratio between $\mathbf{a}_\theta$ and $\mathbf{a}_{\text{safe}}$.

## 3.3 SAFETY CORRECTION MODULE

To prevent the system from violating safety constraints, i.e., leaving $\mathcal{S}_f$, **AutoSafe** incorporates a Safety Correction Module (SCM) that generates a corrective safe action $\mathbf{a}_{\text{safe}} = \mathbf{K}\mathbf{s}_t$, keeping the system within the safety envelope. This design follows the principle of using a verifiable model-based policy to safeguard a learning-based policy (Sha; Zhong et al., 2023; Alshiekh et al., 2018), where a binary switch mechanism typically selects between the two policies based on a risk measure.

Instead of a hard switch, which introduces non-smooth behavior, we employ a continuous weighting scheme. Specifically, we define $\lambda(\mathbf{s})$ as an exponential ramp activation that smoothly increases from 0 to 1 as the system approaches the safety boundary. When operating well inside the safety set $\mathcal{S}_c$, $\lambda(\mathbf{s})$ remains close to 0, giving priority to the learning-based action for higher performance. Near the boundary, $\lambda(\mathbf{s})$ grows toward 1, gradually shifting control authority to the safe policy to ensure that the system stays within the safety set $\mathcal{S}_f$, not violating the safety constraints.

## 3.4 EXPONENTIAL RAMPING ACTIVATION

The introduced Exponential Ramping Activation (ERA), $\lambda(\mathbf{s})$, is defined as a normalized exponential function with enforced boundary conditions at $z(\mathbf{s}) = 0$ and $z(\mathbf{s}) = 1$, such that $\lambda(z(\mathbf{s}) = 0, T) = 0$ and $\lambda(z(\mathbf{s}) = 1, T) = 1$. Its formal expression is given by

$$\lambda(\mathbf{s}) = \frac{e^{z(\mathbf{s})\cdot T} - 1}{e^T - 1}. \tag{5}$$

Here, $T$ is a temperature parameter that controls the sharpness of the exponential growth. A higher $T$ yields a steeper increase of $\lambda$ near the boundary: for most states $\mathbf{s}$, $\lambda(\mathbf{s})$ remains close to 0 and rises sharply to 1 only when approaching the boundary.

The parameter $T$ can be set by a neural network and jointly optimized with the policy network in *actor–critic*-based DRL algorithms (Haarnoja et al., 2018; Schulman et al., 2017). In this formulation, $T$ is updated via gradient ascent together with the policy parameters. After training, the learned $T$ regulates how strongly the safe action contributes to the final control decision, conditioned on the observed information $(\mathbf{s}_t, \mathbf{i}_t)$. In our implementation, $T$ is realized as an additional prediction head of the policy network. Analogously, ERA functions like a human driver who continuously observes the environment and decides when and how strongly to brake in the presence of an imminent hazard. The enforced boundary condition $\lambda(z(\mathbf{s}_t) = 1, T_t) = 1$ guarantees that the safe policy fully takes over when the system reaches the safety boundary.

## 4 INTEGRATION INTO DRL ALGORITHMS

Let us denote **AutoSafe** as the policy $\pi_\lambda$, parameterized by the neural network weights $\theta$. In the combined action output $\mathbf{a}_\lambda = (1 - \lambda(\mathbf{s})) \cdot \mathbf{a}_\theta(\mathbf{o}) + \lambda(\mathbf{s}) \cdot \mathbf{a}_{\text{safe}}(\mathbf{s})$, the stochastic action $\mathbf{a}_\theta \sim \pi_\theta(\cdot \mid \mathbf{o})$ is sampled from a Gaussian policy $\pi_\theta(\mathbf{a} \mid \mathbf{o}) = \mathcal{N}(\mu_\theta(\mathbf{o}), \Sigma_\theta(\mathbf{o}))$, while the safe action $\mathbf{a}_{\text{safe}}$ is computed deterministically as $\mathbf{a}_{\text{safe}} = \mathbf{K}\mathbf{s}$. The weighting parameter $\lambda(\mathbf{s}) \in [0, 1]$ is produced by a learnable exponential function in the actor network.

This construction induces the hybrid policy $\pi_\lambda(\mathbf{a}_\lambda \mid \mathbf{o})$, which can be interpreted as a reparameterized Gaussian policy Cheng et al. (2019a):

$$\pi_\lambda(\mathbf{a}_\lambda \mid \mathbf{o}) = \mathcal{N}(\mu_\lambda(\mathbf{o}), \boldsymbol{\Sigma}_\lambda(\mathbf{o})),$$

with

$$\mu_\lambda(\mathbf{o}) = (1 - \lambda(\mathbf{s}))\mathbf{a}_\theta(\mathbf{o}) + \lambda(\mathbf{s})\mathbf{a}_{\text{safe}}(\mathbf{s}), \quad \boldsymbol{\Sigma}_\lambda(\mathbf{o}) = (1 - \lambda(\mathbf{s}))^2 \boldsymbol{\Sigma}_\theta(\mathbf{o}).$$

Because $\pi_\lambda$ remains a valid and differentiable probability distribution, actions can be sampled and log-likelihoods computed consistently. As a result, **AutoSafe** can be seamlessly integrated into state-of-the-art actor–critic reinforcement learning algorithms, such as SAC and PPO (Haarnoja et al., 2018; Schulman et al., 2017). In this work, we adopt the off-policy SAC algorithm (Haarnoja et al., 2018) for policy learning, where the policy parameters $\theta$ are optimized via gradient ascent on the objective

$$J_{\text{SAC}}(\theta) = \mathbb{E}_{\mathbf{o}_t \sim \mathcal{D}, \, \mathbf{a}_{\lambda_t} \sim \pi_\lambda}[Q_\psi(\mathbf{o}_t, \mathbf{a}_{\lambda_t}) - \alpha \log \pi_\lambda(\mathbf{a}_{\lambda_t} \mid \mathbf{o}_t)], \tag{6}$$

where $Q_\psi$ is the critic network that estimates the expected return when taking action $\mathbf{a}_{\lambda_t}$ given observation $\mathbf{o}_t$, and $\alpha$ is the temperature parameter that balances the entropy bonus.

Since the safety matrix $\mathbf{P}$ and safe control law $\mathbf{K}$ are parameterized as standard matrices and are computed offline, we could implement the whole neural policy as a single neural network (see Figure 2) using widely used machine learning frameworks such as TensorFlow (Abadi et al., 2016) and PyTorch (Ansel et al., 2024). This enables efficient training on parallel computing platforms and flexible deployment in real-world settings, as demonstrated in section 6.3.

## 5 THEORETICAL ANALYSIS

### 5.1 SAFETY PROPERTIES

In our architecture, the safety assurance is fundamentally provided by the model-based safe policy. In the presence of risk, **AutoSafe** effectively constraints the potentially unsafe learning-based actions and keeps the combined action output in the vicinity of the safe ones, such that the safe behavior of the model-based policy is retained for the combined actor $\pi_\lambda$. In fact, the learning-based action $\mathbf{a}_\theta$ is regulated toward the safe action $\mathbf{a}_{safe}$ and the strength of the regulation is controlled by $1 - \lambda(\mathbf{s})$. To analyze the safety regulation property, we leverage the concept of Lyapunov Stability (Basar et al., 2020).

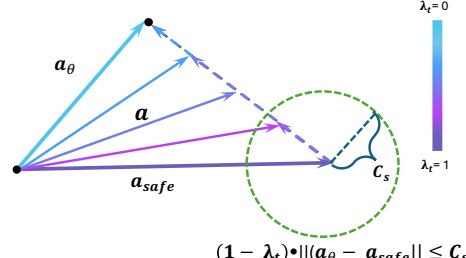

Figure 3: Geometry illustration of action regulation for learning-based action $\mathbf{a}_\theta$

Let us consider a candidate Lyapunov function

$$V(\mathbf{s}) = \mathbf{s}^\top \mathbf{P} \mathbf{s},$$

with $\mathbf{P} \succ 0$, where $\mathbf{P}$ are solved via LMIs in section 2.2. By design, the safe policy $\pi_{\text{safe}}$ asymptotically stabilizes the linear system within $\mathcal{S}_c$, i.e., $\dot{V}(\mathbf{s}) < 0$ for all $\mathbf{s} \in \mathcal{S}_c$ (see proof in section A). Although this Lyapunov function is usually constructed for the linearized system rather than nonlinear dynamics, it still be used as a powerful tool for ensuring safety. In particular, under a moderate assumption of the model mismatch $\mathbf{f}(\mathbf{s}, \mathbf{a})$, we can employ it to study the behavior of the nonlinear system under $\pi_\lambda$, which is stated in the following lemma.

**Lemma 5.1** (Safe Action Regulation). *Assume the unknown model mismatch is bounded as* $\|\mathbf{f}(\mathbf{s}, \mathbf{a})\| \leq \|\mathbf{w}\|$ *for any* $\mathbf{s} \in \mathcal{S}_c$ *and consider the action output* $\mathbf{a}_\lambda = (1 - \lambda)\mathbf{a}_\theta + \lambda\mathbf{a}_{\text{safe}}$; *then, the nonlinear system behaves like a safely controlled linear system (evolving along the direction of decreasing* $V(\mathbf{s})$, *i.e., moving towards the center of the safety envelope) if:*

$$\forall \mathbf{s} \in \mathcal{S}_c, \quad (1 - \lambda(\mathbf{s})) \cdot \|\mathbf{a}_\theta - \mathbf{a}_{\text{safe}}\| < \underbrace{\frac{-\mathbf{s}^\top \mathbf{P} \mathbf{A} \mathbf{s} - \|\mathbf{s}^\top \mathbf{P}\| \cdot \|\mathbf{w}\| - \mathbf{s}^\top \mathbf{P} \mathbf{B} \mathbf{K} \mathbf{s}}{\|\mathbf{s}^\top \mathbf{P} \mathbf{B}\|}}_{C_s}. \tag{7}$$

*Proof.* See section B. □

As illustrated in Figure 3, in the case of $\lambda(\mathbf{s}) \neq 1$, whenever the deviation of $\mathbf{a}_\theta$ from $\mathbf{a}_{\text{safe}}$ remains below $C_s/(1 - \lambda(\mathbf{s}))$, the system evolves inward. The higher is $\lambda$, the easier it is for eq. (7) to hold. When $\lambda(\mathbf{s}) = 1$, the policy reduces exactly to $\pi_{\text{safe}}$, and the condition of eq. (7) reduces to $C_s > 0$. Unfortunately, it is non-trivial to test condition of eq. (7) at each point $\mathbf{s}$ of the boundary $\partial \mathcal{S}_c$; hence, it is common practice in safety critical applications to account for a modest model mismatch by simply shrinking the size of the safety envelope $\mathcal{S}_c$ (see discussion in Sha (2001)). Then, it is straightforward to notice that the policy reduces to $\pi_{\text{safe}}$ at the boundary $\partial \mathcal{S}_c$, which drives the system to move inward. Thus, the system remains within $\mathcal{S}_c \subseteq \mathcal{S}_f$ for all time.

## 5.2 PERFORMANCE ANALYSIS

Prior analyses of hybrid policy frameworks often assume that suboptimal model-based safe policies inevitably introduce bias into the converged solution, thereby degrading performance (Cheng et al., 2019a; Tian et al., 2024). Under this view, achieving equivalent performance would require the contribution of the model-based policy to vanish at convergence. In contrast, our analysis shows that this assumption is unnecessarily restrictive: even though $\pi_{\text{safe}}$ is suboptimal, the policy $\pi_{rl}$ can still converge without losing performance.

Let the value function $V^\pi(\mathbf{s}_0)$ denote the expected return of a policy $\pi$ starting from the initial state $\mathbf{s}_0$. The performance gap between the hybrid policy $\pi_\lambda$ and a standard DRL policy $\pi_{rl}$ can be established by extending the *Performance Difference Lemma* (Kakade and Langford, 2002) as:

$$V^{\pi_\lambda}(\mathbf{s}_0) - V^{\pi_{rl}}(\mathbf{s}_0) = \frac{1}{1 - \gamma} \mathbb{E}_{\mathbf{s} \sim d_{\mathbf{s}_0}^{\pi_\lambda}} \left[ \sum_{\mathbf{a}} (\pi_\lambda(\mathbf{a} \mid \mathbf{s}) - \pi_{rl}(\mathbf{a} \mid \mathbf{s})) Q^{\pi_{rl}}(\mathbf{s}, \mathbf{a}) \right]. \tag{8}$$

In light of eq. (8), at state $\mathbf{s}$, if the action distributions of $\pi_\lambda$ and $\pi_{rl}$ align, i.e., $\pi_\lambda(\mathbf{a} \mid \mathbf{s}) = \pi_{rl}(\mathbf{a} \mid \mathbf{s})$, $\forall a \in \mathcal{A}_f$, then the performance difference does not exists. Within the safety envelope, $\pi_\lambda$ and $\pi_{rl}$ share the same action space. The policy $\pi_\lambda$ can be treated as a parameterized policy family that includes the standard policy $\pi_{rl}$. In the case $\lambda = 1$, the policy $\pi_\lambda$ reduces to $\pi_{rl}$. Moreover, since both $\pi_\lambda$ and $\pi_{rl}$ are optimized under the same learning objective, they admit the same optimal solution. Assume there exists an unique optimal policy $\pi^\star \in \arg\max_\pi J(\pi)$ whose induced visitation measure remains within the safety envelope $\mathcal{S}_c$, i.e., $\Pr_{\pi^\star}[\forall t : \mathbf{s}_t \in \mathcal{S}_c] = 1$. Under this non-binding constraint condition and with sufficient training, both $\pi_\lambda$ and $\pi_{rl}$ can converge to the same optimal policy $\pi^\star$ (Kakade and Langford, 2002; Sutton and Barto, 2018). If multiple optimal policies exist, $\pi_\lambda$ and $\pi_{rl}$ may converge to optimal policies, while achieving the same performance. The derivation of eq. (8) and the extended study on the performance gap are summarized in section C.

## 6 EXPERIMENTS AND RESULTS

We evaluate our method against representative model-based and model-free safe RL baselines on a suite of continuous control tasks ranging from low- to high-dimensional settings.

**Model-Based Baselines.** **Simplex** (Alshiekh et al., 2018; Phan et al., 2020; Cai et al., 2025) (runtime safety shielding by overriding unsafe actions); **AdaLam** (Cheng et al., 2019a; Tian et al., 2024) (weighted summation of safe policy prior and DRL policy, where the weights are adaptively adjusted based on context and safety); **Residual** (Johannink et al., 2019; Cao et al.) (learns a residual policy on top of a fixed safe policy prior);

**Model-Free Baselines.** **Lyapunov** (Westenbroek et al., 2022; Cao et al.) (adds Lyapunov-based penalties for reward shaping for safe exploration). **Lagrangian** (Ha et al., 2020; Achiam et al., 2017; Zhao et al., 2025) (constrained policy optimization via dual variables); **SAC** (Haarnoja et al., 2018) (standard off-policy soft actor–critic algorithm without safety mechanisms);

**Applications**

- **Cartpole Balancing** (Towers et al., 2024): Balance a pole at target position $\hat{x}$ by applying a continuous force input $\mathbf{a} \in \mathbb{R}$ to the cart. Safety constraints: cart position $|x_t| \leq x_{\text{lim}}$ and pole angle $|\theta_t| \leq \theta_{\text{lim}}$.

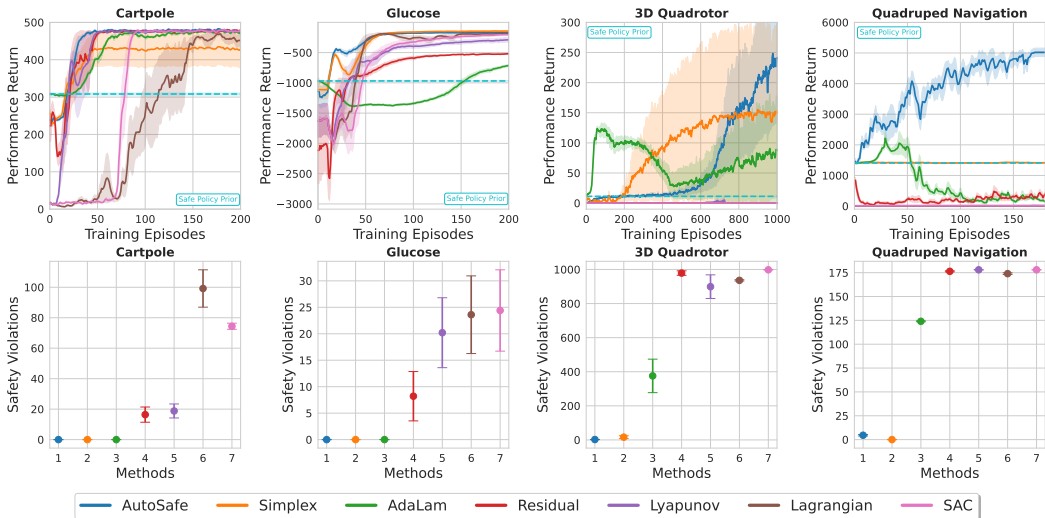

Figure 4: Performance return curves and total safety violations for all methods across 4 continuous control tasks, averaged over 5 random seeds. Shaded regions indicate the standard error of the mean.

- **Glucose Regulation** (Tian et al., 2024): Regulate glucose level $g_t$ using continuous insulin input $\mathbf{a} \in \mathbb{R}$. Safety constraint: $|g_t| \leq g_{\text{lim}}$.
- **3D Quadrotor Goal Reaching** (Yuan et al., 2022): Reach target $\hat{\mathbf{x}} \in \mathbb{R}^3$ using four thrust inputs $\mathbf{a} \in \mathbb{R}^4$. Safety constraints: position $\mathbf{x}_t \leq \mathbf{x}_{\text{lim}}$ and attitude $\boldsymbol{\Theta}_t \leq \boldsymbol{\Theta}_{\text{lim}}$.
- **Quadruped Navigation on Uneven Terrain** (Cao et al.; Yang et al., 2022a): Navigate toward a target zone on uneven terrain using six-dimensional acceleration $\mathbf{a} \in \mathbb{R}^6$. Safety constraints: $z_{\text{min}} \leq z_t \leq z_{\text{max}}$ and $\dot{x}_{\text{min}} \leq \dot{x}_t \leq \dot{x}_{\text{max}}$ (forward movement without falling).

All tasks are formulated as infinite-horizon MDPs. The safety violation condition is identical to the task-specific safety constraints. Once a safety constraint is violated, the episode terminates. We implement all baselines and our method on top of the Vanilla SAC framework (Haarnoja et al., 2018), with further implementation details provided in section D. We run five random seeds for each algorithm on each test case to obtain results.

## 6.1 RESULTS

As shown in Figure 4, **AutoSafe** consistently outperforms all baselines across all tasks, achieving higher returns with almost negligible safety violations. In relatively low-dimensional settings such as Cartpole and Glucose, the shielding-based approach **Simplex** performs comparably to our method; however, its performance degrades substantially in high-dimensional, highly dynamical tasks such as Quadrotor and Quadruped. In these challenging environments, frequent interventions by the safe policy create discontinuities in the data distribution, meaning the state–action transitions suddenly shift from those produced by the learning policy to those imposed by the safe controller. This mismatch makes the data less smooth and harder for the neural network to approximate, causing unstable or diverging learning behavior (see Figure 9 in section E).

When compared to **AdaLam**, which also adaptively updates the parameter $\lambda$ to fuse the learning-based and safe actions, several important differences emerge. In AdaLam, $\lambda$ is typically initialized close to 1, thereby relying heavily on the safe policy for exploration, and then gradually decayed as training progresses. We analyzed three update strategies for $\lambda$, including linear, exponential, and learning-based (see Figure 9 in section E). Our results show that once $\lambda$ decreases, the combined action rapidly loses the safety protection provided by the model-based policy, leading to more frequent safety violations, especially in Quadrotor and Quadruped cases. Conversely, if $\lambda$ remains close to 1 for a long period, the heavy reliance on the safe policy excessively constrains exploration, resulting in inefficient learning and degraded performance. In particular, we found that the learning-based update strategy is ineffective for updating $\lambda$ by maximizing the performance. A further issue is that, since $\lambda$

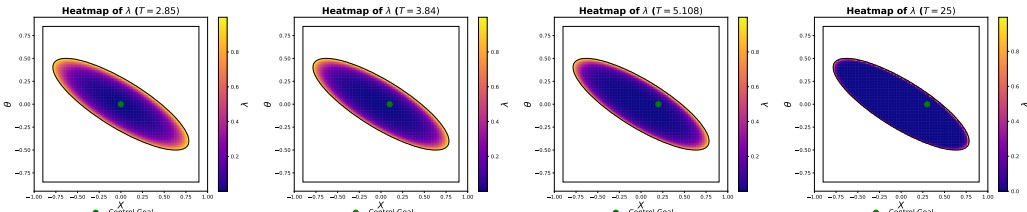

Figure 5: Visualization of $\lambda(\mathbf{s})$ within the safety envelope, shown as slices over the $x$ and $\theta$ dimensions, under different learned values of $T$ for various Cartpole tasks.

and the safe policy are external to the policy (treated as part of the environment), changes in $\lambda$ induce non-stationarities to MDP. To mitigate this, we included $\lambda$ as part of the agent's observation, making the induced dynamics shift observable; however, we found that this adjustment was still insufficient to stabilize learning.

**Residual Learning** can be viewed as a simplified version of **AdaLam**, where the learning-based and safe actions are combined through direct summation. This approach is effective and flexible in relatively simple tasks such as Cartpole and Glucose. However, a common drawback is that the model-based policy may introduce bias, leading to degraded performance, as observed in the Glucose case. Furthermore, if the model-based policy conflicts with the learning objective, strong disagreement between the two policies can arise, which hinders learning, as observed in the Quadruped Navigation task. These observations suggest that the success of the residual architecture relies on the quality of the model-based control prior.

Overall, model-based approaches generally outperform model-free methods in both performance and safety, owing to the incorporation of domain knowledge. In contrast, model-free approaches rarely achieve meaningful progress within the same training time budget, particularly on high-frequency and highly unstable systems. We also observed some failure cases of **AutoSafe**. In the Quadruped task, occasional safety violations occurred during the early stage of training, when learning is unstable due to the limited amount of data in the replay buffer. This instability often results in overly aggressive robot behaviors. We found that introducing a short warm-up period before training helps mitigate such safety violations.

## 6.2 STUDY ON LEARNED $T$

Making the temperature parameter $T$ learnable allows the agent to adaptively adjust its value to maximize performance across different system dynamics and tasks. To validate this hypothesis, we visualize the evolution of $T$ during training (see Figure 12 at section E) and observe that the converged values differ notably across applications. We further design a series of cartpole balancing tasks by intentionally shifting the control goal away from the equilibrium point of the model-based policy, thereby reducing the effectiveness of the safe policy. As the performance of the model-based policy deteriorates, $T$ increases, which in turn reduces $\lambda$. This demonstrates that the agent learns to rely less on a low-performing model-based policy (see Figure 5).

We also explored the other heuristic-based design for the temperature parameter $T$. In particular, we make $T$ schedulable by gradually increasing it to place more weight on the RL policy. We ablate this design using linear and exponential schedules (see Figure 10 at section E). We found that the learning-based approach consistently performs well across all applications without task-specific tuning. However, the scheduled approach requires task-specific adjustments and potentially introduces design bias, as seen in the Quadrotor case in Figure 10 at section E.

## 6.3 REAL WORLD APPLICABILITY

The proposed architecture is particularly suitable for safe online policy learning in real-world settings, which could be helpful to bridge domain gaps such as sim-to-real and offline-to-online reinforcement learning for continuous control tasks. We demonstrate this capability in a real-world continual learning task on the cartpole, shown in Figure 6a, where the **AutoSafe** policy is deployed on an embedded device (Raspberry Pi) for safe interaction, while its weights are periodically updated from

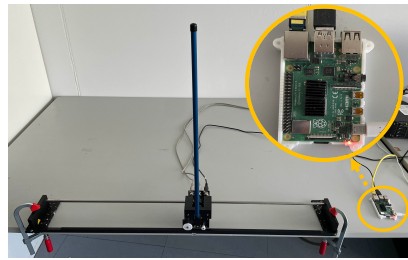
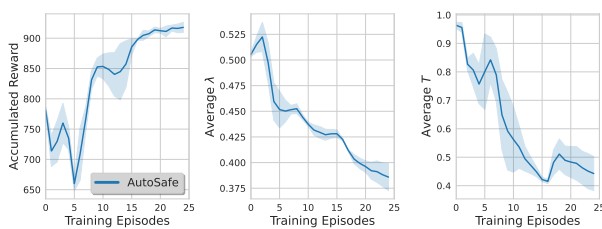

(a) AutoSafe deployed on an embedded device (Raspberry Pi 4B) for safe interaction.

(b) Experimental results from real-world deployment.

Figure 6: Real-world experiments with **AutoSafe**. The **AutoSafe** agent is deployed on an embedded device for safe interaction at the frequency of 50hz. The weights of the **AutoSafe** are periodically updated from a policy optimized on a remote workstation.

a remote workstation. As illustrated in Figure 6b, **AutoSafe** enables the agent to interact safely with the environment while continually improving its performance. Interestingly, we observe that the learned $T$ decreases over time to a smaller value, producing a smoother exponential ramp. This potentially suggests that the agent adapts by favoring smoother actions to cope with the stochasticity of real-world dynamics, since abrupt action changes can induce large disturbances and degrade stabilization performance.

## 7 RELATED WORK

Recent advancements, such as constrained policy optimization(Wachi and Sui, 2020; Achiam et al., 2017), policy-prior aided training(Xie et al., 2018), and Lyapunov-reward designs(Westenbroek et al., 2022; Zhao et al., 2023; Cao et al.) have improved safety during training and incentivized the controlled agent to learn to be safe. Despite these advances, formal safety guarantees during deployment remain unclear.

Alternatively, some methods employ a separate safe policy that backs up the DRL policy when safety conditions are violated (Zhong et al., 2023; Phan et al., 2020). However, this approach suffers from several issues: the safe policy and DRL policy are designed separately, leading to overly aggressive DRL actions that frequently trigger safety overrides, and the mechanism for switching back to the DRL policy remains understudied. Another line of work is the verification of the learned deep policies (Yang et al., 2024; Wang et al., 2021). However, the verification is typically done offline with model assumptions and is often not ready to be transferred to real-world applications.

A promising direction is to integrate a certified safe model-based policy (Sha, 2001) with the DRL policy, allowing them to work collaboratively to achieve near-optimal performance while maintaining runtime safety. While recent frameworks using residual learning (Johannink et al., 2019), regularization (Cheng et al., 2019a), or policy fusion (Rana et al., 2023) have shown promising results, they focus on improving data efficiency and performance during training rather than enforcing safety constraints to main safety. Moreover, they often rely on heuristically selected or data-estimated parameters to combine the policy, which often sacrifices the verifiability of the model-based safe policy and compromises the performance.

## 8 CONCLUSIONS AND LIMITATIONS

This paper proposes **AutoSafe**, a novel neural policy architecture that enables DRL agents to learn safely under various safety constraints, achieving superior performance compared to existing safe RL methods. However, the current constraints are assumed to be addressed by using a uniform safety envelope and safe policy. In a real-world scenario, the constraint might be multi-layered, e.g, a humanoid robot would need to avoid obstacles while walking stably. In this case, we would need to have a hierarchical safety layer design to satisfy all safety constraints at the same time. How to efficiently address all safety constraints at different levels constitutes the future research direction.

## REPRODUCIBILITY STATEMENT

We provide a detailed description of the **AutoSafe** architecture in Section 3, with proofs, assumptions, and additional implementation details and experiments in the appendix. Experimental settings and baseline settings are described in Section 6 and Appendix D. Our source code and scripts are included in the supplementary materials to facilitate replication.

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

# Appendices

## A   COMPUTATION OF SAFETY ENVELOPE AND SAFE POLICY

There are many different ways for model-based safe policy design; we adopt the procedure introduced in Seto and Sha (1999) to search for a safe policy and obtain the safety envelope. In this section, we summarize the general problem formulation and solution. For the detailed implementation and results, we refer the interested readers to the codes attached in the supplementary materials.

Given that a nonlinear system described as

$$\dot{\mathbf{s}} = \underbrace{\mathbf{As} + \mathbf{Ba}}_{known} + \underbrace{\mathbf{f}(\mathbf{s}, \mathbf{a})}_{unknown} , \tag{9}$$

where $\mathbf{s} \in \mathbb{R}^n$ and $\mathbf{a} \in \mathbb{R}^m$. We aim to search for a safe policy that drives the system to operate safely within the safety set $s_t \in \mathcal{S}_f$ characterized by a set of safety constraints in state space.

$$\text{Safety set} : \mathcal{S}_f \coloneqq \{\mathbf{s} \in \mathbb{R}^n | \mathbf{A}_s \mathbf{s} \le \mathbf{b}_s\}. \tag{10}$$

under the action constraints $\mathcal{A}_f \coloneqq \{\mathbf{a} \in \mathbb{R}^m | \mathbf{A}_a \mathbf{a} \le \mathbf{b}_a\}$. $\mathbf{A}_s \in \mathbb{R}^{n_s \times n}$, $\mathbf{b}_s \in \mathbb{R}^{n_s}$ represent $n_s$ polytopic/ellipsoidal constraints on the state space. $\mathbf{A}_a \in \mathbb{R}^{n_a \times m}$, $\mathbf{b}_a \in \mathbb{R}^{n_a}$, representing $n_a$ number of the action constraints. The safety requirement is formally stated as follows:

**Definition A.1** (Safety Definition). *Consider the safety set $\mathcal{S}_f$ eq. (10). The system eq. (9)is said to be safe, if given any $\mathbf{s}_t \in \mathcal{S}_f$, the $\mathbf{s}_{t+1} \in \mathcal{S}_f$ holds for any time $t \in \mathbb{N}$.*

To design a verifiable model-based safe policy directly for the non-linear system eq. (9) is difficult due to the unknown terms $\mathbf{f}(\mathbf{s}, \mathbf{a})$. Instead, we consider a time-invariant linear approximation of eq. (9) as:

$$\dot{\mathbf{s}} = \mathbf{As} + \mathbf{Ba} \tag{11}$$

We parameterize the model-based safe policy as a matrix $\mathbf{K} \in \mathbb{R}^{m \times n}$, and the action $\mathbf{a}$ can be computed as $\mathbf{a} = \mathbf{Ks}$. The system eq. (11) with the state constraints and action constraints can then be rewritten as

$$\dot{\mathbf{s}} = \overline{\mathbf{A}} \cdot \mathbf{s}, \quad \text{with constraints: } \mathbf{A}_c \mathbf{s} \le \mathbf{1}, \quad \mathbf{A}_c \in \mathbb{R}^{n_c \times n}, \quad n_c = n_s + n_a \tag{12}$$

where $\overline{\mathbf{A}} = \mathbf{A} + \mathbf{BK}$, and $\mathbf{A}_c$ denotes the unified constraint matrix obtained by expressing the action constraints in terms of the state via $\mathbf{a} = \mathbf{Ks}$. To be more specific, $\mathbf{A}_c$ is obtained by stacking the action constraints with the state constraints, and normalizing so that the right-hand side is the all-ones vector $\mathbf{1}$.

The objective is to find a control gain $\mathbf{K}$ such that the closed-loop system in eq. (12) is asymptotically stable. Under this condition, a control-invariant subset of the safety set, the safety envelope (introduced below), is forward invariant, ensuring that trajectories initialized within it remain inside the safety set and thus safe. To establish the relation between the choice of $\mathbf{K}$ and the stability region associated with the control using $\mathbf{K}$, we apply *Lyapunov Stability* analysis.

**Definition A.2.** *Seto and Sha (1999) The system eq. (12) is quadratically stable if there exists a positive definite matrix $\mathbf{P} \succ 0$ such that the quadratic function $V(\mathbf{s}) = \mathbf{s}^\top \mathbf{Ps}$ has negative derivatives along all the trajectories of eq. (12).*

The Lyapunov stability criterion states that the system in eq. (12) is asymptotically stable if and only if it is quadratically stable. Hence, it is sufficient to study a quadratic Lyapunov function. The derivative of the function $V$ is

$$\dot{V}(\mathbf{s}) = \mathbf{s}^\top (\overline{\mathbf{A}}^\top \mathbf{P} + \mathbf{P}\overline{\mathbf{A}})\mathbf{s} \tag{13}$$

We can conclude that the system eq. (12) is asymptotically stable if and only if there exists a matrix $\mathbf{P}$ such that

$$\mathbf{P} \succ 0, \quad \overline{\mathbf{A}}^\top \mathbf{P} + \mathbf{P}\overline{\mathbf{A}} \preceq 0 \quad \text{or} \quad \mathbf{Q} = \mathbf{P}^{-1} \succ 0, \quad \mathbf{Q}\overline{\mathbf{A}}^\top + \overline{\mathbf{A}}\mathbf{Q} \preceq 0 \tag{14}$$

With $\mathbf{P}$, the stability region $\mathcal{S}_\mathbf{c}$ can be defined as

$$\text{Stability region} : \mathcal{S}_\mathbf{c} \triangleq \{\mathbf{s} \in \mathbb{R}^n | \mathbf{s}^\top \mathbf{Ps} \le 1\}. \tag{15}$$

Apparently, the stability region $\mathcal{S}_\mathbf{c}$ is a subset of the safety set eq. (10), every point inside the region satisfies the constraints $\mathbf{A}_c \mathbf{s} \le \mathbf{1}$, with each row $\alpha_k^\top \mathbf{s} < 1, k = 1, \ldots c$, which impose a constraint on matrix $\mathbf{P}$, formally stated as follows:

**Lemma A.3.** *Seto and Sha (1999) Given A LTI system with constraints in eq.* (12). *The stability region $\mathcal{S}_{\mathbf{c}}$ of eq.* (12) *satisfies the constraint in eq.* (12) *if and only if $\alpha_k^\top \mathbf{P}^{-1} \alpha_k \leq \mathbf{1}$ The size of the stability region is proportional to $\sqrt{\det\left(\mathbf{P}^{-1}\right)}$*

*Proof.* The proof can be found in Seto and Sha (1999), appendix A2. □

The solution of $P$ is not unique; we are interested in deriving the largest stability region, which can be formulated as a linear matrix inequality (LMI) problem as:

$$\underset{\mathbf{Q},\mathbf{K}}{\text{minimize}} \quad \log \det \mathbf{Q}^{-1} \tag{16}$$

$$\text{subject to} \quad \mathbf{Q}\mathbf{A}^\top + \mathbf{A}\mathbf{Q} + \mathbf{Q}\mathbf{K}^\top\mathbf{B}^\top + \mathbf{B}\mathbf{K}\mathbf{Q} \leq 0 \tag{17}$$

$$\mathbf{Q} \succ 0 \tag{18}$$

$$\alpha_k^\top \mathbf{Q} \alpha_k \leq 1 \tag{19}$$

$$\begin{bmatrix} \mathbf{I} & \mathbf{b}_j^\top\mathbf{K}\mathbf{Q} \\ \mathbf{Q}^\top\mathbf{K}^\top\mathbf{b_j} & \mathbf{Q} \end{bmatrix} \geq 0, j = 1, ..., r \tag{20}$$

The constraint eq. (17) is derived by substituting $\overline{\mathbf{A}} = \mathbf{A} + \mathbf{B}\mathbf{K}$ in eq. (14). eq. (19) is the conclusion from theorem A.3. eq. (20) is the conclusion of theorem A.3 applied on the action constrain, where $\mathbf{b}_j^\top\mathbf{a} \leq 1, j = 0, \ldots n_a$ is the normized constraint vector at each row of $\mathbf{A}_a$. The expression is obtained by $\mathbf{b_j}^\top\mathbf{a} = \mathbf{b}_j^\top\mathbf{K}\mathbf{s} \leq 1 \Rightarrow \mathbf{b}_j^\top\mathbf{K}\mathbf{Q}\mathbf{K}^\top\mathbf{b_j} \leq 1$, then converted to the LMI formulation using *Schur-complement*Zhang (2006). The optimization problem can be solved using CVX toolbox Grant et al. (2009).

## B  SAFE ACTION REGULATION LEMMA

**Lemma B.1** (Safe Action Regulation). *Assume the unknown model mismatch is bounded as* $\|\mathbf{f}(\mathbf{s}, \mathbf{a})\| \leq \|\mathbf{w}\|$ *for any* $\mathbf{s} \in \mathcal{S}_c$ *and consider the action output* $\mathbf{a}_\lambda = (1 - \lambda)\mathbf{a}_\theta + \lambda \mathbf{a}_{safe}$; *then, the nonlinear system behaves like a safely controlled linear system (evolving along the direction of decreasing* $V(\mathbf{s})$, *i.e., moving towards the center of the safety envelope) if:*

$$\forall \mathbf{s} \in \mathcal{S}_c, \quad (1 - \lambda(\mathbf{s})) \cdot \|\mathbf{a}_\theta - \mathbf{a}_{\text{safe}}\| < \underbrace{\frac{-\mathbf{s}^\top \mathbf{PAs} - \|\mathbf{s}^\top \mathbf{P}\| \cdot \|\mathbf{w}\| - \mathbf{s}^\top \mathbf{PBKs}}{\|\mathbf{s}^\top \mathbf{PB}\|}}_{C_s}. \tag{21}$$

*Proof.* From section A, we obtain a true Lyapunov function $V(\mathbf{s})$ for the linear system eq. (11). The obtained control law $\mathbf{K}$ renders the region $\mathcal{S}_c$ *forward invariant*. For the real system eq. (3), if the system is stable, the derivative of the Lyapunov function $\dot{V}(\mathbf{s}) < 0$.

$$\begin{aligned}
\dot{V}(\mathbf{s}) &= 2\mathbf{s}^\top \mathbf{P}\dot{\mathbf{s}} \\
&= 2\mathbf{s}^\top \mathbf{P}(\mathbf{As} + \mathbf{Ba} + \mathbf{f}(\mathbf{s}, \mathbf{a})) \\
&= 2\mathbf{s}^\top \mathbf{P}\left(\mathbf{As} + \mathbf{B}(\mathbf{a}_\theta + \lambda(\mathbf{s})(\mathbf{a}_{\text{safe}} - \mathbf{a}_\theta)) + \mathbf{f}(\mathbf{s}, \mathbf{a})\right) \\
&= 2\mathbf{s}^\top \mathbf{PAs} + 2\mathbf{s}^\top \mathbf{PB}(\mathbf{a}_\theta + \lambda(\mathbf{s})(\mathbf{a}_{\text{safe}} - \mathbf{a}_\theta)) + 2\mathbf{s}^\top \mathbf{Pf}(\mathbf{s}, \mathbf{a})
\end{aligned} \tag{22}$$

If the system is stable $\dot{V}(\mathbf{s}) < 0$, in light from eq. (22), we can get

$$\mathbf{s}^\top \mathbf{PAs} + \mathbf{s}^\top \mathbf{PB}(\mathbf{a}_\theta + \lambda(\mathbf{s})(\mathbf{a}_{\text{safe}} - \mathbf{a}_\theta)) + \mathbf{s}^\top \mathbf{Pf}(\mathbf{s}, \mathbf{a}) < 0$$
$$\mathbf{s}^\top \mathbf{PB}(\mathbf{a}_\theta + \lambda(\mathbf{s})(\mathbf{a}_{\text{safe}} - \mathbf{a}_\theta)) + \mathbf{s}^\top \mathbf{Pf}(\mathbf{s}, \mathbf{a}) < -\mathbf{s}^\top \mathbf{PAs} \tag{23}$$

Assume that the model mismatch $\mathbf{f}(\mathbf{s}, \mathbf{a})$ is bounded, i.e, $\|\mathbf{f}(\mathbf{s}, \mathbf{a})\| \leq \|\mathbf{w}\|$ for all $\mathbf{s} \in S_c, \mathbf{a} \in \mathcal{A}_f$, according to the *Cauchy-Schwarz inequality*, we can get the upper-bound of the term $\mathbf{s}^\top \mathbf{Pf}(\mathbf{s}, \mathbf{a})$ as

$$\mathbf{s}^\top \mathbf{Pf}(\mathbf{s}, \mathbf{a}) \leq \|\mathbf{s}^\top \mathbf{P}\| \cdot \|\mathbf{f}(\mathbf{s}, \mathbf{a})\| \leq \|\mathbf{s}^\top \mathbf{P}\| \cdot \|\mathbf{w}\| \tag{24}$$

Therefore eq. (23) can be converted as:

$$\mathbf{s}^\top \mathbf{PB}(\mathbf{a}_\theta + \lambda(\mathbf{s})(\mathbf{a}_{\text{safe}} - \mathbf{a}_\theta)) + \|\mathbf{s}^\top \mathbf{P}\| \cdot \|\mathbf{w}\| < -\mathbf{s}^\top \mathbf{PAs}$$
$$\mathbf{s}^\top \mathbf{PB}(\mathbf{a}_\theta + \lambda(\mathbf{s})(\mathbf{a}_{\text{safe}} - \mathbf{a}_\theta)) < -\mathbf{s}^\top \mathbf{PAs} - \|\mathbf{s}^\top \mathbf{P}\| \cdot \|\mathbf{w}\|$$
$$\tag{25}$$

Let's introduce a variable $\beta(\mathbf{s}) = (1 - \lambda(\mathbf{s}))$, eq. (23) can be converted as:

$$\mathbf{s}^\top \mathbf{PB}(\mathbf{a}_\theta + (1 - \beta(\mathbf{s})(\mathbf{a}_{\text{safe}} - \mathbf{a}_\theta)) < -\mathbf{s}^\top \mathbf{PAs} - \|\mathbf{s}^\top \mathbf{P}\| \cdot \|\mathbf{w}\|$$
$$\mathbf{s}^\top \mathbf{PB}(\mathbf{a}_{\text{safe}} + \beta(\mathbf{s})(\mathbf{a}_\theta - \mathbf{a}_{\text{safe}})) < -\mathbf{s}^\top \mathbf{PAs} - \|\mathbf{s}^\top \mathbf{P}\| \cdot \|\mathbf{w}\|$$
$$\beta(\mathbf{s}) \cdot \mathbf{s}^\top \mathbf{PB}(\mathbf{a}_\theta - \mathbf{a}_{\text{safe}}) < -\mathbf{s}^\top \mathbf{PAs} - \|\mathbf{s}^\top \mathbf{P}\| \cdot \|\mathbf{w}\| - \mathbf{s}^\top \mathbf{PBKs}$$
$$\tag{26}$$

Apply *Cauchy-Schwarz inequality* to the left side, we get:

$$\beta(\mathbf{s}) \cdot \|\mathbf{s}^\top \mathbf{PB}\| \cdot \|\mathbf{a}_\theta - \mathbf{a}_{\text{safe}}\| < -\mathbf{s}^\top \mathbf{PAs} - \|\mathbf{s}^\top \mathbf{P}\| \cdot \|\mathbf{w}\| - \mathbf{s}^\top \mathbf{PBKs}$$

$$\beta(\mathbf{s}) \cdot \|\mathbf{a}_\theta - \mathbf{a}_{\text{safe}}\| < \frac{-\mathbf{s}^\top \mathbf{PAs} - \|\mathbf{s}^\top \mathbf{P}\| \cdot \|\mathbf{w}\| - \mathbf{s}^\top \mathbf{PBKs}}{\|\mathbf{s}^\top \mathbf{PB}\|}$$

$$(1 - \lambda(\mathbf{s})) \cdot \|\mathbf{a}_\theta - \mathbf{a}_{\text{safe}}\| < \frac{-\mathbf{s}^\top \mathbf{PAs} - \|\mathbf{s}^\top \mathbf{P}\| \cdot \|\mathbf{w}\| - \mathbf{s}^\top \mathbf{PBKs}}{\|\mathbf{s}^\top \mathbf{PB}\|}$$

$$\tag{27}$$

$\square$

## C  PERFORMANCE ANALYSIS

We leverage the *Performance Difference Lemma* Kakade and Langford (2002) to prove the performance difference analysis between the standard DRL policy and **AutoSafe**.

**Lemma C.1.** *Performance Difference Lemma* Kakade and Langford (2002)
*The performance difference $\Delta V$ for any two policies $\pi_1$ and $\pi_2$ at any state $\mathbf{s}_0 \in \mathcal{S}$ is:*

$$\Delta V = V^{\pi_1}(\mathbf{s}_0) - V^{\pi_2}(\mathbf{s}_0) = \frac{1}{1-\gamma} \mathbb{E}_{\mathbf{s} \sim d_{\mathbf{s}_0}^{\pi_1}} \Big[ \mathbb{E}_{\mathbf{a} \sim \pi_1(\cdot|\mathbf{s})} [A^{\pi_2}(\mathbf{s}, \mathbf{a})] \Big], \tag{28}$$

*where $A^{\pi_2}(\mathbf{s}, \mathbf{a}) = Q^{\pi_2}(\mathbf{s}, \mathbf{a}) - V^{\pi_2}(\mathbf{s})$, $d_{\mathbf{s}_0}^{\pi}$ is the discounted occupancy of state $\mathbf{s}$ starting from $\mathbf{s}_0$ by following $\pi_1$.*

Continuing from the theorem C.1, let's expand the advantage function $A$, and we can get the following:

$$V^{\pi_1}(\mathbf{s}_0) - V^{\pi_2}(\mathbf{s}_0) = \frac{1}{1-\gamma} \mathbb{E}_{\mathbf{s} \sim d_{\mathbf{s}_0}^{\pi_1}} \Big[ \mathbb{E}_{a_1 \sim \pi_1(\cdot|\mathbf{s})} [Q^{\pi_2}(\mathbf{s}, \mathbf{a}_1) - V^{\pi_2}(\mathbf{s})] \Big]$$

$$= \frac{1}{1-\gamma} \mathbb{E}_{\mathbf{s} \sim d_{\mathbf{s}_0}^{\pi_1}} \Big[ \mathbb{E}_{\mathbf{a}_1 \sim \pi_1(\cdot|\mathbf{s})} [Q^{\pi_2}(\mathbf{s}, \mathbf{a}_1)] - \mathbb{E}_{\mathbf{a}_2 \sim \pi_2(\cdot|\mathbf{s})} [Q^{\pi_2}(\mathbf{s}, \mathbf{a}_2)] \Big] \tag{29}$$

$$= \frac{1}{1-\gamma} \mathbb{E}_{\mathbf{s} \sim d_{\mathbf{s}_0}^{\pi_1}} \Big[ \sum_{\mathbf{a}} (\pi_1(\mathbf{a} \mid \mathbf{s}) - \pi_2(\mathbf{a} \mid \mathbf{s})) Q^{\pi_2}(\mathbf{s}, \mathbf{a}) \Big] \tag{30}$$

Suppose a standard DRL policy $\pi$ without the safe action can converge to a standard RL policy: $\pi_{k=\infty} = \pi_{rl}$. Let's consider the performance gap for the policy $\pi_\lambda$ and the standard RL policy $\pi_{rl}$: $V^{\pi_\lambda}(\mathbf{s}) - V^{\pi_{rl}}(\mathbf{s})$. When $\lambda(\mathbf{s}) = 1$, $\mathbf{a}_\lambda(\mathbf{s}) = (1-1) \cdot \mathbf{a}_\theta + 1 \cdot \mathbf{a}_{\text{safe}} = \mathbf{a}_{\text{safe}}$, effectively only the safe policy is interacting with the environment, therefore $V^{\pi_\lambda}(\mathbf{s}) - V^{\pi_{rl}}(\mathbf{s}) = V^{\pi_{\text{safe}}}(\mathbf{s}) - V^{\pi_{rl}}(\mathbf{s})$. When $\lambda(\mathbf{s}) = 0$, $\mathbf{a}_\lambda(\mathbf{s}) = (1-0) \cdot \mathbf{a}_\theta + 0 \cdot \mathbf{a}_{\text{safe}} = \mathbf{a}_\theta$, only the learning based policy is interacting with the environment. Under the assumption that a standard DRL policy can converge to the standard RL policy $\pi_{\theta_{k=\infty}} = \pi_{rl}$, the performance gap is $V^{\pi_\lambda}(\mathbf{s}) - V^{\pi_{rl}}(\mathbf{s}) = V^{\pi_\theta}(\mathbf{s}) - V^{\pi_{rl}}(\mathbf{s}) = 0$. It is more interesting to investigate the performance gap when $\lambda(\mathbf{s}) \neq 0$, detailed below.

In light of eq. (30), we have

$$V^{\pi_\lambda}(\mathbf{s}_0) - V^{\pi_{rl}}(\mathbf{s}_0) = \frac{1}{1-\gamma} \mathbb{E}_{\mathbf{s} \sim d_{\mathbf{s}_0}^{\pi_\lambda}} \Big[ \sum_a (\pi_\lambda(\mathbf{a} \mid \mathbf{s}) - \pi_{rl}(\mathbf{a} \mid \mathbf{s})) Q^{\pi_{rl}}(\mathbf{s}, \mathbf{a}) \Big]$$

$$\tag{31}$$

### C.1  PERFORMANCE GAP UPPER BOUND

Directly analyzing the impact of $\lambda(\mathbf{s})$ on the performance is not trackable due to the nonlinearity and nonconvexity of the value function eq. (8). Instead, we analyze the upper bound of the performance gap to give some insights into the influences of the parameter $\lambda(\mathbf{s})$. In light of eq. (8), we have:

$$\big|V^{\pi_\lambda}(\mathbf{s}_0) - V^{\pi_{rl}}(\mathbf{s}_0)\big| = \frac{1}{1-\gamma} \left| \mathbb{E}_{\mathbf{s} \sim d_{\mathbf{s}_0}^{\pi_\lambda}} \Big[ \sum_{\mathbf{a}} \big(\pi_\lambda(\mathbf{a} \mid \mathbf{s}) - \pi_{rl}(\mathbf{a} \mid \mathbf{s})\big) Q^{\pi_{rl}}(\mathbf{s}, \mathbf{a}) \Big] \right|$$

$$\leq \frac{1}{1-\gamma} \mathbb{E}_{\mathbf{s} \sim d_{\mathbf{s}_0}^{\pi_\lambda}} \Big[ \sum_{\mathbf{a}} \big|\pi_\lambda(\mathbf{a} \mid \mathbf{s}) - \pi_{rl}(\mathbf{a} \mid \mathbf{s})\big| \big|Q^{\pi_{rl}}(\mathbf{s}, \mathbf{a})\big| \Big]$$

$$\leq \frac{1}{1-\gamma} \mathbb{E}_{\mathbf{s} \sim d_{\mathbf{s}_0}^{\pi_\lambda}} \Big[ Q_{\max}^{\pi_{rl}}(\mathbf{s}) \sum_{\mathbf{a}} \big|\pi_\lambda(\mathbf{a} \mid \mathbf{s}) - \pi_{rl}(\mathbf{a} \mid \mathbf{s})\big| \Big]$$

$$= \frac{1}{1-\gamma} \mathbb{E}_{\mathbf{s} \sim d_{\mathbf{s}_0}^{\pi_\lambda}} \Big[ Q_{\max}^{\pi_{rl}}(\mathbf{s}) \big\|\pi_\lambda(\cdot \mid \mathbf{s}) - \pi_{rl}(\cdot \mid \mathbf{s})\big\|_1 \Big] \tag{32}$$

$$= \frac{2}{1-\gamma} \mathbb{E}_{\mathbf{s} \sim d_{\mathbf{s}_0}^{\pi_\lambda}} \Big[ Q_{\max}^{\pi_{rl}}(\mathbf{s}) \, \text{TV}\big(\pi_\lambda(\cdot \mid \mathbf{s}), \pi_{rl}(\cdot \mid \mathbf{s})\big) \Big]. \tag{33}$$

where eq. (32) is obtained by applying the $L_1$ norm on two distributions; eq. (33) is obtained by converting the $L_1$ using *total-variations distance* (**TV**).

# D EXPERIMENTAL DETAILS

## D.1 ALGORITHM IMPLEMENTATIONS

### D.1.1 SOFT ACTOR-CRITIC (SAC) BASELINE

Our implementation of Soft Actor-Critic follows (Haarnoja et al., 2018) and (Fujimoto et al., 2018), with default parameters summarized in Table 2. These parameters are shared across all baseline algorithms unless specified otherwise.

Table 1: Parameter setting for Soft Actor-Critic (SAC)

| Hyperparameter | Value |
|---|---|
| Discount factor ($\gamma$) | 0.99 |
| Learning rate (actor, critic) | $3 \times 10^{-4}$ |
| Optimizer | Adam |
| Target smoothing coefficient ($\tau$) | 0.005 |
| Entropy coefficient | 0.1 |
| Target update interval | 1 |
| Activation function | ReLU |
| Training steps per environment step | 1 |
| Evaluation period (steps) | 10000 |
| Neural Network (MLP) | [256, 256] |
| Batch size | 128 |

Table 2: Task Specific Parameter Setting

| Hyperparameter | CartPole | Glucose | Quadrotor | Quadruped |
|---|---|---|---|---|
| Replay buffer size | $2 * 10^5$ | $1 * 10^5$ | $1 * 10^6$ | $5 * 10^6$ |
| Entropy coefficient | 0.1 | 0.2 | 0.005 | 0.005 |
| Total training steps | $2 * 10^5$ | $1 * 10^5$ | $1 * 10^6$ | $5 * 10^5$ |
| Maximum steps per episode | 500 | 200 | 1000 | 3000 |

### D.1.2 AUTOSAFE

The proposed **AutoSafe** neural policy additionally has a temperature prediction head compared to the standard actor network in SAC. This prediction head shares the same backbone as the action prediction head and is activated using the tanh activation function. We then apply an affine transformation to map the output range $(-1, 1)$ into $(T_{\min}, T_{\max})$. We set $T_{\min} = 1.0$ and $T_{\max} = 25.0$ for all studied case studies. The selection of the range can be flexible. We set $T_{\max} = 25.0$ to ensure the DRL agent has enough freedom to explore the state using its own action within the safety envelope. The value of $\lambda$ is capped to 1 for the case of exceeding safety envelope caused by abrupt environmental uncertainties such as disturbance or noises.

### D.1.3 SIMPLEX

The implementation of the Simplex architecture follows the standard setting as in Sha et al. (2001); Phan et al. (2020). The system's safety is backed up by a safe policy. The switching between the safe and learning-based policy is determined by a state-dependent function, detailed as follows:

$$\mathbf{a}_t = \begin{cases} \mathbf{a}_{safe_t} = \pi_{\text{safe}}(\mathbf{s}_t), & \text{if } \mathbf{s}_t^\top \mathbf{P} \mathbf{s} \geq 1, i.e., \ \mathbf{s} \notin \mathcal{S}_c \\ \mathbf{a}_{\theta t} \sim \pi_\theta(\cdot \mid \mathbf{s}_t), & \text{otherwise.} \end{cases}$$

The matrix $\mathbf{P}$ is solved from the LMIs problemA and identical to the $\mathbf{P}$ in **AutoSafe**. The safe policy $\pi_{safe}$ generates the safe action as $\mathbf{a}_{safe} = \mathbf{K}\mathbf{s}$.

## D.2 APPLICATIONS

### D.2.1 CARTPOLE

**Task Definition:** The goal of this task is to balance the pole at the intended target position $\hat{x}$ by driving the cart using force input. In this task, the observation of the agent is defined as $o_t = \{x_t, \dot{x}_t, \sin(\theta), \cos(\theta), \dot{\theta}\}$. Our method and the other methods that use a model-based safe prior require the tracking error $e_t$ as the additional input for the safe policy to generate a safe action. $e_t$ is defined as the difference between the current state $\mathbf{s}_t$ and the control equilibrium $\mathbf{s}^*$ of the model-based design, $e_t = \{x_t - x^*, \dot{x} - \dot{x}^*, \theta_t - \theta^*, \dot{\theta}_t - \dot{\theta}^*\}$. The equilibrium is set as $\mathbf{s}^* = \{0, 0, 0, 0\}$. The control loop is running at 50Hz.

We append the tracking error $e_t$ to the observation space of the other model-free methods to ensure all algorithms receive the same amount of information. We adopt the reward function proposed in Cao et al. (2022), formulated as

$$r = e^{-\delta \cdot d(\mathbf{s}_t - \hat{\mathbf{s}})} - \beta \cdot \mathbf{a}_t^2,$$

where $\delta = 5$ is a parameter that adjusts the smoothness of the exponential function; $d(\cdot)$ is the Euclidean distance between the current state $\mathbf{s}_t$ and the control target $\hat{\mathbf{s}} = \{0.1, 0, 0, 0\}$; $\beta$ is a parameter to balance the reward and the action penalty. For more details about the task setting, we refer interested readers to Cao et al. (2022).

**Safety Constraints:** In this task, the safety constraints are

$$|x_t| \leq x_{\text{lim}} \quad |\theta_t| \leq \theta_{\text{lim}},$$

where $x_{\text{lim}} = 0.5$ m and $\theta_{\text{lim}} = 0.785$ rad.

**Model Based Design** The safety envelope and safe policy are obtained from solving an LMI problem, as discussed in section A. The system model can be found at Florian (2007). The linearized model and the code to calculate the matrix $\mathbf{P}$ and $\mathbf{F}$ are available in the attached supplementary files. For more details of solving LMIs for the cartpole task, we refer the interested reader to Cao et al.; Seto and Sha (1999).

### D.2.2 GLUCOSE

**Task Definition** The problem setting and simulation of this application are adopted from Tian et al. (2024). In the blood glucose regulation task, the goal is to regulate the blood glucose level $G$ to minimize the Magni risk Fox et al. (2020) by controlling insulin injection $a_I$. The dynamics of the glucose control problem are governed by the following ODEs Tian et al. (2024),

$$\dot{G} = -p_1(G - G_b) - GX + D_t,$$
$$\dot{X} = -p_2 X + p_3(I - I_b),$$
$$\dot{I} = -n(I - I_b) + a_I$$

Here, $G$ represents the amount of glucose in the blood, and $I$ represents the amount of insulin in the blood. $X$ describes the delayed effect of insulin on lowering blood glucose, which is often unobservable. In this task, the observation is the $o_t = \{G_t, \Delta G_t, t\}$, where $\Delta G_t = G_t - G_{t-1}$ and $t$ is the total time passed after meal ingestion. The equilibrium of the model-based design is set to be the normal fasting level of glucose and insulin. $\mathbf{s}^* = \{G^*, X^*, I^*\} = \{138, 0, 7\}$. The reward function is defined as

$$r = \begin{cases} -(3.35506 \times ((\ln^{0.8353}) - 3.7932)^2 & \text{if } 10 \leq G \leq 1000, \\ -1e3 & \text{otherwise} . \end{cases}$$

**Safety Constraints** We follow the safety constraints introduced in Tian et al. (2024) as:

$$G_{\text{min}} \leq G_t \leq G_{\text{max}},$$

where $G_{\text{min}} = 10$ and $G_{\text{max}} = 1000$.

**Moded-based Design** The safety envelope and safe policy are obtained from solving an LMI problem, as discussed in section A. The system model can be found at Tian et al. (2024). The linearized model and the code to calculate the matrix $\mathbf{P}$ and $\mathbf{K}$ are available in the attached supplementary files.

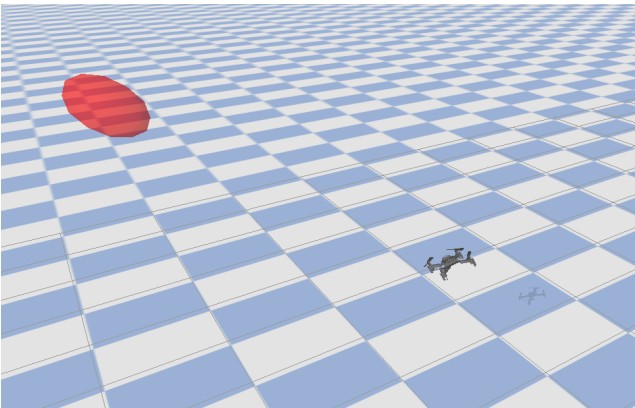

Figure 7: Setup of quadrotor goal reaching. The red sphere represents the target zone
.

### D.2.3 3D QUADROTOR GOAL REACHING

**Task Definition** The goal of this task is to control a quadrotor to reach a target goal position $\{\hat{x}, \hat{y}, \hat{z}\}$ by controlling the thrust input on each propeller. The observation for the agent is $o = \{x, y, z, \theta_x, \theta_y, \theta_z, \dot{x}, \dot{y}, \dot{z}, \dot{\theta}_x, \dot{\theta}_y, \dot{\theta}_z\}$. The action space consists of a four-dimensional trust input, denoted as $a = \{u_1, u_2, u_3, u_4\}$. The equilibrium of the model-based design is $s^* = \{0, 0, 0, 0, 0, 0, 0, 0, 0, 0, 0, 0, 0\}$. The reward function is adopted from Yuan et al. (2022) as:

$$r = e^{-\alpha \cdot (\|x-\hat{x}\|^2 + \|y-\hat{y}\|^2 + \|z-\hat{z}\|^2) - \beta \cdot \|a\|^2},$$

where $\alpha = 1.0$ and $\beta = 1e-4$ are the weights to balance the distance-related reward and action penalty. In our case study, we set the initial position of the quadrotor as $\mathbf{s}_{0_{xyz}} = \{1.5, 1.5, 1.5\}$ and the target position of the quadrotor as $\hat{\mathbf{s}}_{xyz} = \{2.5, 2.5, 2.5\}$. The control loop is running at 50Hz.

**Safety Constraints** The safety constraint is defined as

$$x_{\min} < x_t < x_{\max}, \quad y_{\min} < y_t < y_{\max}, \quad z_{\min} < z_t < z_{\max} \tag{34}$$

where $x_{\min} = -5.0 \text{ m}$, $y_{\min} = -5.0 \text{ m}$, $z_{\min} = 0.0 \text{ m}$, and $x_{\max} = y_{\max} = z_{\max} = 5.0 \text{ m}$, representing the allowable moving area in the $x - y - z$ space.

**Model-based Design** The safety envelope and safe policy are obtained from solving an LMI problem, as discussed in section A. The system model can be found at Yuan et al. (2022). The linearized model and the code to calculate the matrix $\mathbf{P}$ and $\mathbf{F}$ are available in the attached supplementary files.

### D.2.4 QUADRUPED NAVIGATION ON UNEVEN TERRAIN

**Task Definition** In this task, we aim to train a safe RL policy to enable the quadruped robot to walk through the uneven terrain to finish a virtual package delivery problem. The robot needs to first go to a package pick-up zone (A) and then navigate a package drop zone (B). We assume a virtual package is automatically attached to the robot when the robot reaches zone A and detached when it reaches zone B. The terrain is created unevenly by randomly placing blocks on the floor, with the maximum height of the unevenness to be 4 centimeters. The control loop is running at 200Hz.

The observation of the agent includes the pose of the robot in the world coordinates and the relative distance toward goal A for picking up, and the relative distance toward goal B for dropping off. We additionally add the task phase ID, $i$ (0 or 1), for the picking up and delivery. Overall, $o$ is defined as $o = \{x, y, z, \theta_x, \theta_y, \theta_z, \dot{x}, \dot{y}, \dot{z}, \dot{\theta}_x, \dot{\theta}_y, \dot{\theta}_z, x_{rel}, y_{rel}, z_{rel}, i\}$. The action space is defined as the desired acceleration as $a = \{\ddot{x}, \ddot{y}, \ddot{z}, \ddot{\theta}_x, \ddot{\theta}_y, \ddot{\theta}_z\}$. The acceleration input is then mapped to the low-level joint angles using a Model Predictive Controller (MPC) (Yang et al., 2022b). The equilibrium point for the model-based design is defined as $s^* = \{0, 0, 0.24, 0, 0, 0, 0.26, 0, 0, 0, 0, 0, 0\}$, meaning maintaining the height $z = 0.24$ and forward velocity $\dot{x} = 0.26$ m/s. The reward is designed piece-wise to incentivise the robot to move to A and B to finish the whole task, as

$$r(\mathbf{s}) = w_d \, r_d(\mathbf{s}, g) + w_h \, r_h(\mathbf{s}, g) + w_z \, r_z(\mathbf{s}) + R_1 \, \mathbb{I}_{\text{pickup}} + R_2 \, \mathbb{I}_{\text{delivery}}.$$

Figure 8: Setup of quadruped navigation. The target pick-up zone is indicated as a blue spot, and the delivery zone is indicated as a yellow spot. The terrain is randomized with uneven height.

Each term of the reward is defined as:

$$\text{positional reward}: r_d(\mathbf{s}, g) = -\|\mathbf{s}_{xyz} - \hat{\mathbf{s}}_{xyz}\|,$$
$$\text{directional reward}: r_h(\mathbf{s}, g) = -|\theta - \theta_g|,$$
$$\text{height regulation reward}: r_z(\mathbf{s}) = -|z - z_{\text{ref}}|,$$

$$\mathbb{I}_{\text{pickup}} = \begin{cases} 1, & \text{if reaches the pickup zone (A)} \\ 0, & \text{otherwise} \end{cases}$$

$$\mathbb{I}_{\text{delivery}} = \begin{cases} 1, & \text{if reaches the delivery zone (B)} \\ 0, & \text{otherwise} \end{cases}$$

where the coefficient of each term is detailed as

$$w_d = 1.0, \quad w_h = 0.25, \quad w_z = 0.25 \quad R_1 = 2.5, \quad R_2 = 50.$$

**Safety Constraints** The safety constraint in this task is mainly considered as the robot not falling and jumping too high while moving forward, as:

$$z_{\min} \le z_t \le z_{\max}$$

where $z_{\min} = 0.16$ m and $z_{\max} = 0.8$ m.

**Model-based design** The safety envelope and safe policy are obtained from solving an LMI problem, as discussed in section A. The system model can be found at Cai et al. (2025). The linearized model and the code to calculate the matrix $\mathbf{P}$ and $\mathbf{F}$ are available in the attached supplementary files.

# E ADDITIONAL EXPERIMENTAL RESULTS

## E.1 DIVERGENCE OF THE SIMPLEX-BASED METHODS

We visualize the training loss curves for **AutoSafe** and **Simplex**. We observe that training of the **Simplex** is gradually diverging with a large critic loss, as shown in Figure 9. We attribute this to the frequent interventions by the safe policy, which creates discontinuities in the data distribution, meaning the state–action transitions suddenly shift from those produced by the learning policy to those imposed by the safe controller. This mismatch makes the data less smooth and harder for the neural network to approximate, causing unstable or diverging learning behavior.

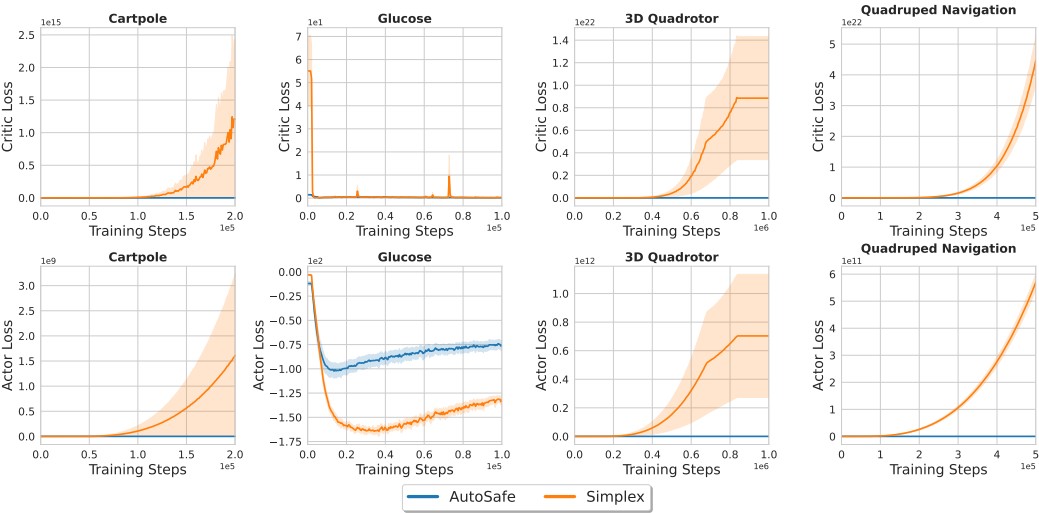

Figure 9: Critic Loss and Actor Loss of our and Simplex-Based Method

## E.2 ABLATION ON THE SETTING OF TEMPERATURE $T$

In this case study, we compare our learnable temperature setting against two heuristics, including linear and exponential increasing. The result is shown in fig. 10. We found that all methods work similarly regarding the performance, except for the quadrotor case. However, designing an effective scheduling scheme is non-trivial, which requires repetitive manual tuning. Moreover, the manual setting might introduce bias.

## E.3 ABLATION OF ADALAM

In this study, we investigate several ways of setting the weights between safe action and learning-based action, as shown in fig. 11. We found out that initializing the $\lambda$ to close to 1 at the beginning of the training enables safe interactions. For simple tasks, such as cartpole and glucose, the agent could learn using the data generated by the safe policy. However, we found that it is not effective in high-dimensional cases. For a learning-based setting, the exploration is not effective; therefore, the performance of the policy is barely improved. For the scheduled setting, we found that frequent safety violations occur when $\lambda$ decreases. The frequent safety violations generate a lot of uninformative data, where the agent cannot learn to converge.

## E.4 EVOLUTION OF TEMPERATURE PARAMETER

In this section, we visualize the evolution of the learnable temperature parameter $T$ during the policy learning. It can be seen that the learned temperature converges to the different value given different tasks, which suggests that it might be not trivial to manually set the "right" parameter using heuristics.

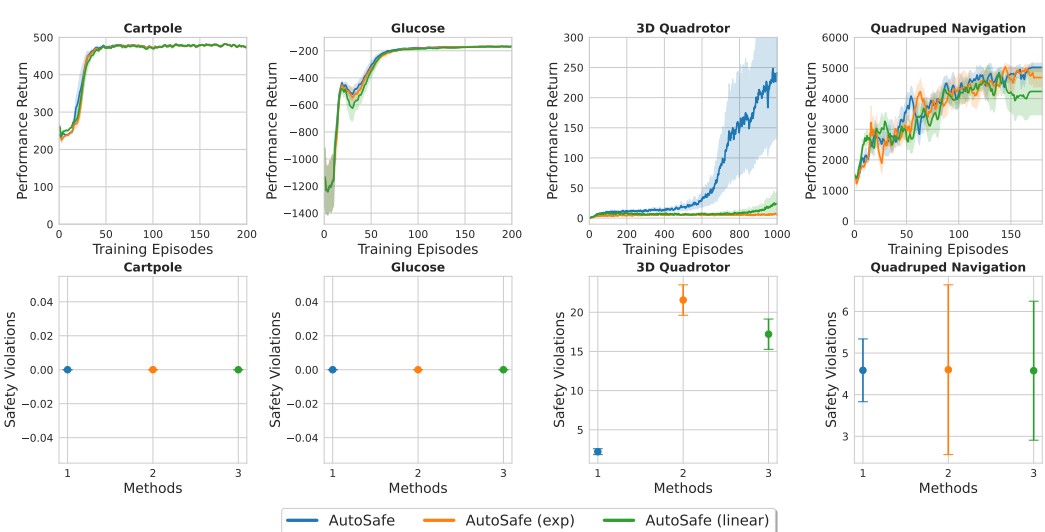

Figure 10: Ablation Study of temperature parameter $T$: Learning-based vs. Schedule-based.

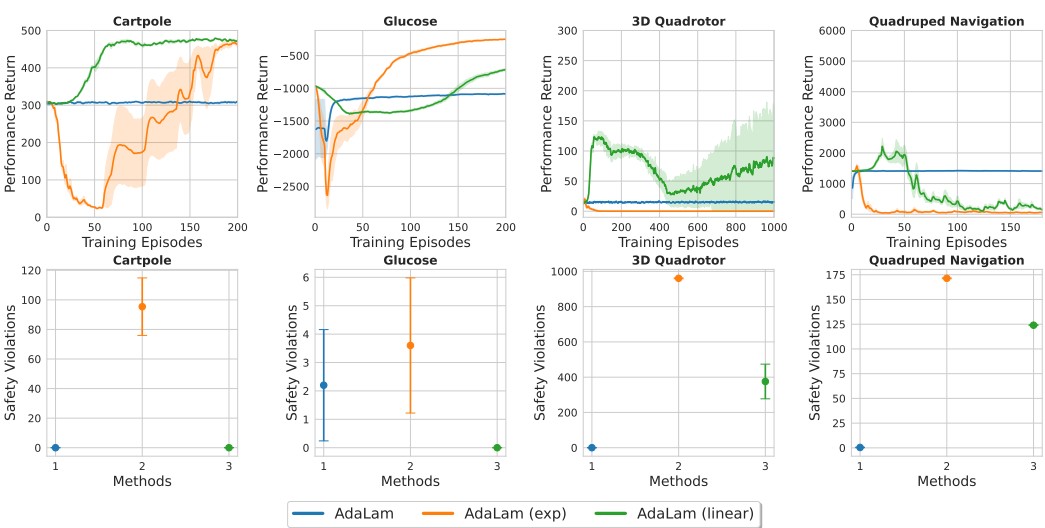

Figure 11: AdaLam

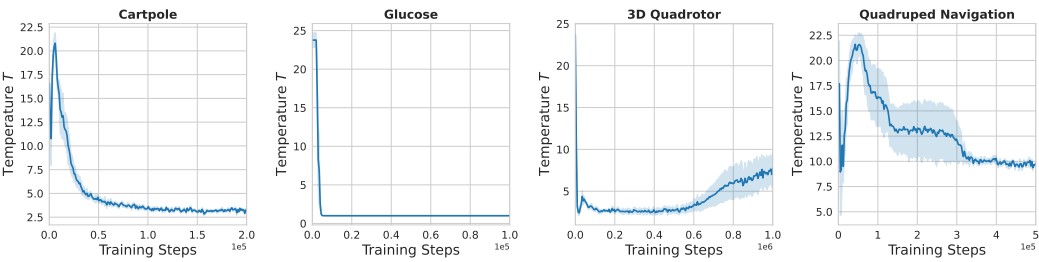

Figure 12: The evolution of temperature parameter $T$ during policy learning for studied applications

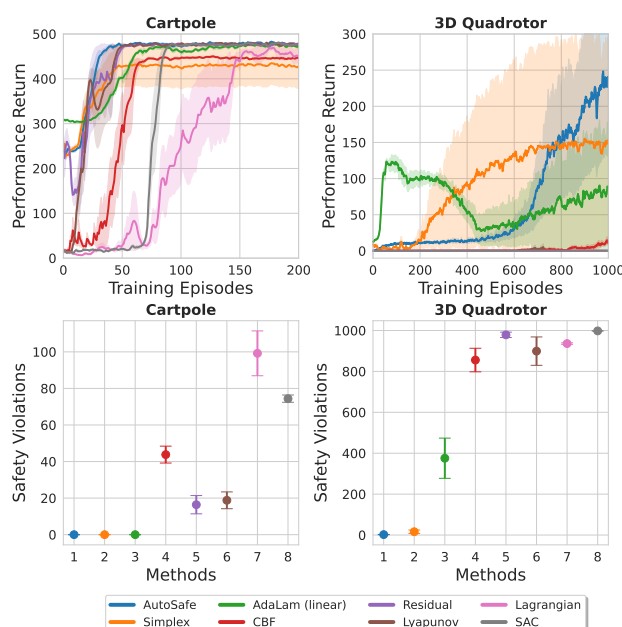

Figure 13: CBF vs. the other methods

## E.5 RESULTS OF CONTROL-BARRIER FUNCTION ON CARTPOLE AND QUADROTOR

In this experiment, we implement a Control-Barrier Function based approach for the safety back up mechanism. Our implementation of CBF follows the setup introduced in Ames et al. (2019). In particular, given an action from a DRL policy $\mathbf{a}_\theta$, a CBF-based safety filter computes a safe action $\mathbf{a}_{safe}$ by solving the following quadratic program:

$$\mathbf{a}_{safe} = \arg\min_{\mathbf{a}} \frac{1}{2}\|\mathbf{a} - \mathbf{a}_\theta\|^2 \tag{35}$$

$$\text{s.t.} \quad L_f h(\mathbf{s}) + L_g h(\mathbf{s})\mathbf{a} \geq -\alpha(h(\mathbf{s})) \tag{36}$$

$$\mathbf{A}_a \mathbf{a} \leq \mathbf{b}_a, \tag{37}$$

where $\alpha(h(\mathbf{s})) = k \cdot h(\mathbf{s})$ and $h(\mathbf{s})$ is defined as:

$$h(\mathbf{s}) = \mathbf{b}_s - \mathbf{A}_s \mathbf{s} \quad h(\mathbf{s}) \in \mathbb{R}^{n_s}. \tag{38}$$

Here, $h(\mathbf{s})$ is defined in matrix form, yielding a vector-valued barrier function. In practice, each component of this vector is enforced as an individual CBF constraint. The $(\mathbf{A}_s, \mathbf{b}_s)$ and $(\mathbf{A}_a, \mathbf{b}_a)$ correspond to the state constraints and action constraints, respectively, as introduced in the background section. For the experiments of cartpole and quadrotor we set $k = 0.5$. The results are shown in Figure fig. 13. We observed that the constructed CBF baseline reduces safety violations, but its performance remains below that of the Simplex approach. In practice, we found that solving the CBF QP online is sensitive to parameter choices and can frequently become infeasible, especially near the safety boundary. Additional parameter tuning and system-specific tuning may improve its performance, but this process is nontrivial and beyond the scope of this work.

## E.6 RESULTS OF GOAL CONFLICTING

In this experiment, we design a position tracking problem in cartpole: the agent needs to control the cartpole to reach and balance at the target position. We intentionally make the goal of the safe policy to be at $x_{goal} = 0.0$ m and create a list of goal conflicting scenarios for DRL with

$$x_{goal} = \{0.0, 0.1, 0.2, 0.3\} \text{ m},$$

where the larger number corresponds to the stronger goal conflict. 0.3 is set near the boundary of the safety envelope which has the strongest conflict. As shown in Figure 14, our method can handle

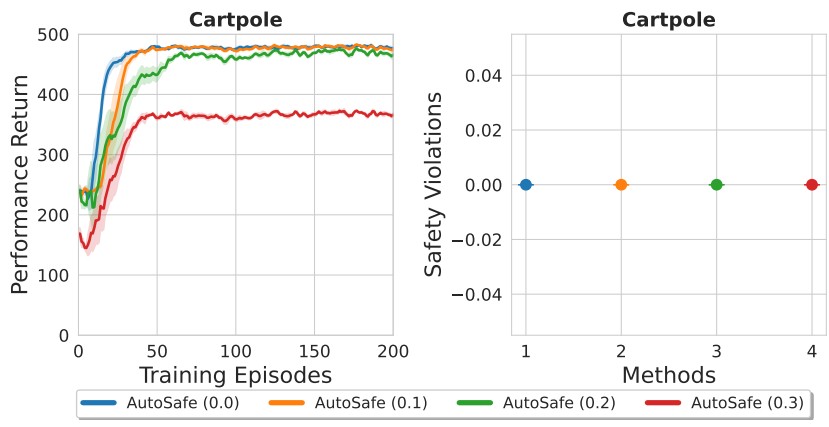

Figure 14: Ablation study on the goal conflicting for **AutoSafe**

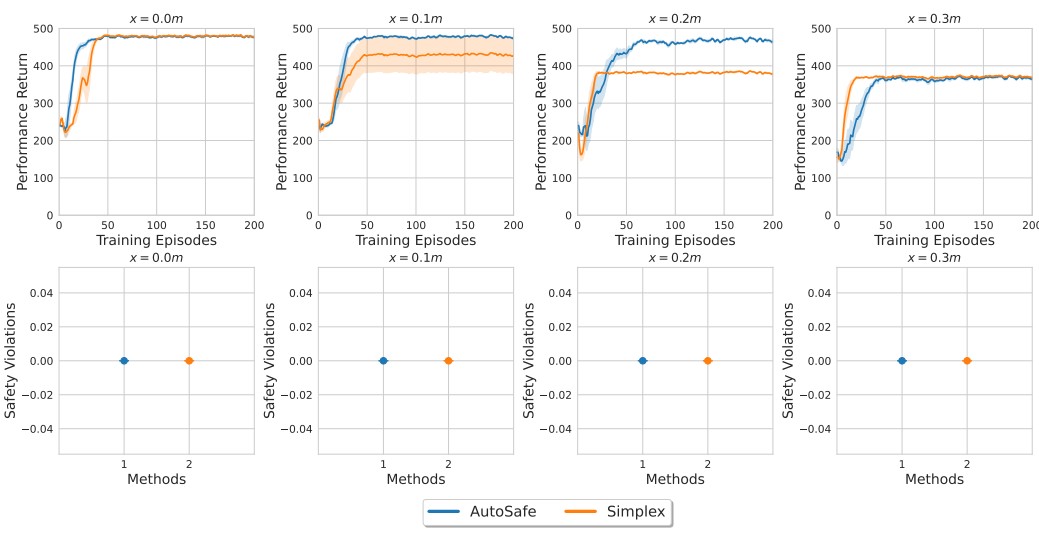

Figure 15: Ablation study on the goal conflicting for **AutoSafe** and **Simplex**

different levels of the goal conflicting, outperforming the *hard-switch* counterpart **Simplex** (see comparison results in Figure 15). However, it can also be noticed that our method also struggles when the DRL goal is very close the safety boundary where the safe policy inevitably intervene to keep the system safe. In such cases, performance is naturally constrained to be degraded compared to the settings with weaker goal conflict.

# F   STATEMENT ON USING LARGE LANGUAGE MODEL (LLM)

We clarify that the LLM is only employed for polishing the writing and for assisting in the generation of the illustrative image in Figure 1, which serves to facilitate the explanation of the mathematical concepts.

