# OpenReview forum: "AutoSafe: A Safe Neural Policy Network with Safety Common Sense"
_ICLR.cc/2026/Conference — Submitted to ICLR 2026_

### Official Review · Reviewer_pSva · 2025-10-19

**Soundness:** 2
**Presentation:** 3
**Contribution:** 2
**Rating:** 2
**Confidence:** 4

**Summary:**

The paper proposes AutoSafe, a safe reinforcement learning framework that blends a learned policy with a model-based safety controller. It constructs a safe action using a Lyapunov function and a safety envelope. The approach seems to yield a differentiable, end-to-end trainable architecture that automatically regulates the trade-off between performance and safety. The paper provides theoretical analysis alongside its experimental evaluation.

**Strengths:**

The strengths of this paper are:
- AutoSafe extends existing Lyapunov-based Safe RL methods in a somewhat novel way. The proposed ERA is novel and neat. While smooth blending between controllers is not new, the ERA’s specific exponential design and learnable temperature seem fresh in this contextual literature, and I appreciate its addition.
- Real-world experiment is a great addition!
- Multiple test-suite envs show good generalisability.

**Weaknesses:**

The weaknesses of this paper are:
- Correct me if I have misunderstood. The safety guarantee of AutoSafe holds if the system dynamics are known/partially known, the model mismatch ||w|| is uniformly bounded (under assumption), and the states are fully observable. For the experiments, how much of the systems dynamics was encoded as a priori? Does the uniform bound of ||w|| hold under real-world dynamics outside of controlled experiments?
- The paper never states how the safety violation is defined? Is it a priori safety constraint? Is it a terminal state?
- I feel like some baselines are missing or could be replaced. See references below, as some examples:
- - Approximate model-based shielding for safe reinforcement learning
- - End-to-End Safe Reinforcement Learning through Barrier Functions
- - Safe Model-based Reinforcement Learning with Stability Guarantees
- - etc
- Novelty seems incremental, and without comparison to stronger baselines, it is hard to gauge genuine strengths which is a shame because I enjoyed the proposed ERA and paper as a whole

**Questions:**

- Could the authors clarify how much prior model knowledge was used for each experimental domain?
- In the real-world robot setting, does the uniform bound on ||w|| realistically hold?
- Could the authors explain why the stronger baselines were excluded, or comment on how AutoSafe would perform relative to these methods (e.g., computational overhead, robustness, and safety tightness)?
- Could the authors elaborate on what aspects they consider fundamentally new relative to prior Lyapunov or barrier-function safety integration approaches outside of ERA?

---

> ### Author Response · Authors · 2025-11-27
> **Response to Weaknesses**
>
> We appreciate your positive feedback on the innovation of ERA design and its validation in real-world applications and various simulated benchmarks. We would like to clarify some details and questions in the following responses.
>
> ### W1: The safety guarantee of AutoSafe holds if the system dynamics are known/partially known, the model mismatch ||w|| is uniformly bounded (under assumption), and the states are fully observable. For the experiments, how much of the system's dynamics was encoded a priori? Does the uniform bound of ||w|| hold under real-world dynamics outside of controlled experiments?
>
> Your understanding is correct. For all experiments, we need to know the linear model approximation of the system, where the model mismatch is considered in the linearization process, as well as the safety constraints. Then the building block safety matrix $P$ and Safe Policy $K$ can be solved using standard LMI tools.
>
> When deployed to a new environment, unfortunately, it is non-trivial to test each state in the state space to know the exact bound. Hence, it is common practice in safety-critical applications to account for a modest model mismatch by simply shrinking the size of the safety envelope Sc (see discussion in Sha (2001)) such that the model mismatch bound still holds for the true system.
>
> ### W2: The paper never states how the safety violation is defined. Is it an a priori safety constraint? Is it a terminal state?
>
> In all studied experiments, the safety violation condition is defined as the same as the prior safety constraints. If the safety constraints are violated, the system terminates. We added the detailed explanation in the experiment setup section of the revised paper.
>
>
> ### W3: I feel like some baselines are missing or could be replaced. See references below, as some examples: .... Novelty seems incremental, and without comparison to stronger baselines, it is hard to gauge genuine strengths
>
> As pointed out by reviewer otqo, the Simplex baseline operates according to the same fundamental principle as classical shielding or CBF-style safety filters:
> The controller continuously checks whether the system is entering an unsafe region, and if so, replaces the learning-based action with a safe one to maintain safety.
> This “monitor–intervene” mechanism is shared across Simplex, CBFs, and other intervention-based safety architectures.[1].
>
> In particular, CBFs rely on solving an online QP using a dynamics model to find an action that satisfies the barrier condition. Simplex switches to an offline-certified backup controller that guarantees a decrease in the Lyapunov function. Despite these procedural differences, the safety intervention principle is essentially the same.
>
> In our experiments, the Simplex baseline indeed demonstrates the strongest empirical safety guarantee, exhibiting the lowest number of safety violations among all methods.
>
> To address the reviewer’s concern, we additionally implemented a CBF baseline for both the Cartpole and Quadrotor environments (results included in **Appendix E**).
> We found that the CBF approach did not perform well in practice. We attribute it to the following reasons:
>
> 1. Online QP does not always yield a valid solution, especially when the learned or nominal policy proposes actions that violate the barrier hard constraints.
> 2. Model mismatch, the CBF relies on an approximate dynamics model for predicting $h(s')$ online, moderate model errors could lead to overly conservative or incorrect safety decisions.
>
> ### W4: Novelty
> We summarize the key novelties of our proposed method as follows:
>
> 1. **A new perspective on safe action backup.**
>    We introduce a *safe action blending* mechanism, which contrasts with existing strategies such as action replacement, action projection, and action masking [2]. This provides a different method for safeguarding the action output of the unverified deep neural network.
>
> 2. **ERA enables smooth and safety-preserving controller blending.**
>    ERA is designed to continuously blend a learning-based controller with a model-based safety controller while maintaining formal safety guarantees. By avoiding hard switching when risk is detected, ERA keeps the training transitions more continuous, which is especially beneficial for online policy learning.
>
> 3. **Extensive evaluation in both simulation and real-world settings.**
>    We demonstrate performance improvements over relevant baselines across multiple simulation benchmarks and further validate the proposed approach in real-world experiments.
>
> [1] Hsu, Kai-Chieh, Haimin Hu, and Jaime F. Fisac. "The safety filter: A unified view of safety-critical control in autonomous systems." Annual Review of Control, Robotics, and Autonomous Systems 7 (2023).
>
> [2] Krasowski, Hanna, et al. "Provably Safe Reinforcement Learning: Conceptual Analysis, Survey, and Benchmarking." Transactions on Machine Learning Research (2023)

---

> ### Author Response · Authors · 2025-11-27
> **Response to Questions**
>
> ### Q1: Could the authors clarify how much prior model knowledge was used for each experimental domain? In the real-world robot setting, does the uniform bound on ||w|| realistically hold?
>
> Please refer to **Response to W1**.
>
> ### Q2: Could the authors explain why the stronger baselines were excluded, or comment on how AutoSafe would perform relative to these methods?
>
> We did not exclude any strong baselines. The implemented Simplex baseline is a representative method for white-box shielding-based approaches, including CBF-based methods, as also noted by reviewer *otqo*. The key differences across these approaches lie in:
>
> - **Which safety function** is used to detect impending violations, and
> - **How the safe action** is generated to enforce safety.
>
> In our experiments, the Simplex baseline indeed provides the strongest empirical assurance of safety. Additional experiments using a CBF-based shield did not yield improved safety or performance. Further discussion is provided in **Response to W3**.
>
> ### Q3: Could the authors elaborate on what aspects they consider fundamentally new relative to prior Lyapunov or barrier-function safety integration approaches outside of ERA?
>
> Our main contribution is an **architectural innovation**. Specifically, ERA unifies the classical safety filter framework [1] with the residual policy architecture [2], enabling a smooth and adaptive transition between the learning-based controller and the model-based safety controller. This provides several benefits:
>
> 1. Unified, application-friendly architecture
> AutoSafe does not require explicit predictive models or online optimization (e.g., solving QPs as in CBF or MPC-based methods).  In real-world applications, AutoSafe outputs safe actions directly in a single forward pass, whereas other methods require coordinating two separate controllers (one for safe and one for learning).
>
> 2. Easier and more stable training
> Compared to the architecture that separates the safety layer and learning-based approach, our architecture is internally safety-aware. During the training, the gradients flow to the learnable parameters to accommodate the safe policy (not only at the boundary). This creates smooth behaviors of the agent, in the sense that the sampled data is more continuous, making the neural network approximation easier and therefore converging faster, which is especially important for real-world learning, where generated samples are expensive. Furthermore, the proposed architecture can be integrated into on-policy algorithm (PPO), however it might be difficult for the other safe architecture as the other safe RL baseline are essentially use two polices to generate rollouts.
>
> [1] Hsu et al. *The Safety Filter: A Unified View of Safety-Critical Control in Autonomous Systems*, 2023.
>
> [2] Johannink, Tobias, et al. *Residual Reinforcement Learning for Robot Control*. ICRA, 2019.

---

### Official Review · Reviewer_otqo · 2025-10-22

**Soundness:** 3
**Presentation:** 3
**Contribution:** 2
**Rating:** 2
**Confidence:** 4

**Summary:**

This paper presents AutoSafe, a SafeRL method designed to train a policy to optimize an objective given constraints.
AutoSafe builds on control theory for systems with linear dynamics and linear constraints to derive a model-based safety test and a safety policy. These are directly integrated in the policy learning through a smooth merging approach. It is shown for SAC in the paper. The method is tested on several environments and against several safe RL baselines, showing better performance. There is also a study about the temperature additional parameter $T$ which is learned.

**Strengths:**

The paper takes part in an important line of work. Building safe RL agents is an important challenge towards real world applications and works in this direction should be encouraged.
The article’s argument is well built. The method is sound under the stated hypotheses, as it allows agents to learn optimal policies while ensuring the respect of constraints. The paper showcases experiments on different environments, including a real world device, showcasing the performance of the method. There are also relevant additional results which allow to better understand the influence of temperature $T$. Finally, the paper is generally easy to read, with an apparent effort to convey the information clearly, although there is some missing information or confusing notation here and there.

**Weaknesses:**

There are three main weaknesses to the paper.

First, the approach is limited to systems for which (1) the dynamics are (approximately) known and (2) the dynamics and the constraints can be expressed in a linear manner so as to compute the exact solution for matrices K and P, needed for the safety-check and safe policy. In itself, this is not a problem because these assumptions can be considered true for several relevant applications. Yet, they should be explicitly stated in the introduction and abstract, as the proposed method does not solve the problem of Safe RL, but it proposes a solution for specific types of problems (known and linear dynamics, and linear constraints). It does not do it through “Safety Common Sense” but by benefitting from the specific assumptions of this type of problem. The paper would benefit from a less shiny, more rigorous packaging.

Second, the contribution itself is not clear to me. Overall, the proposed method is very, very similar to the general framework of (white-box) shielding [1,2]/provably safe RL [3] (specifically, the concept of Control Barrier Functions [4]), where prior knowledge of the world is used to determine if the action is safe and otherwise replace it with a safe action. AutoSafe is integrated into the policy and the policy parameters $\theta$ are learned accordingly, but this seems actually equivalent to the agent learning on an environment where the shield is deployed. The only different is the smoothening between the safe and the parameterized policy, but given a safe policy and a value of lambda (which is conditioned on s and stable once $T$ has converged), the parameterized can learn to compensate for the safe policy bias. The safe policy therefore only matters when $\lambda = 1$; and the approach is once again equivalent to white-box shielding. The fact that $T$ can be learned and change during the learning introduces weird learning dynamics but it is not clear how and why this would be beneficial.

The results show the approach to be better than white-box shielding (called “simplex”). I am not sure why this is the case. The authors refer to “non-stationarities” in the simplex learning, which I don’t understand (white-box shielding is stationary if the shield is conditioned on (s,a), as it does not change during learning). This is the last important weakness of the paper: there are not enough information about the simplex baseline (see questions to the authors) to correctly interpret the results.

[1] Könighofer et al. (2017) Shield synthesis

[2] Alshiekh et al. (2018) Safe reinforcement learning via shielding  (cited by the authors)

[3] Krasowski et al. (2023) Provably Safe Reinforcement Learning: Conceptual Analysis, Survey, and Benchmarking

[4] Hsu et al. (2023) The Safety Filter: A Unified View of Safety-Critical Control in Autonomous Systems

**Questions:**

**Introduction:**
In the second paragraph, it is said that the issue with external safety layers is that they often over constrain the agent’s performance. This can be the case for learned safety layers. Well built shields can be less conservative while having full safety guarantees (a bit like what is done in this paper…) but they need prior knowledge of the environment.

**Section 2.2**
Equation (3): is $\dot{s}$ the time derivative of $s$? While this is conventional in control and dynamics fields, it could be worth defining for the RL/ML community.

**Section 3.1**
Doesn’t the weighted summation approach assume that the action space is “ordered” (for a lack of better word)? In the sense that if $a_1 > a_2$, the effect on $s’ \sim P(s,a)$ is always in the same direction? More concretely: if in $a \in [-1,  0]$ you go to left (-1 max move, 0 no move) and $a \in [0, 1]$ you go right (0 no, 1 max), it works. If inversely between -1 and 0 you go to left, this time with (-1 no, 0 max) and between 0 to 1 you go right (0 more, 1 less), you have a discontinuity in the effect of the action on the environment, and this would completely fail (although it would be a valid MDP).

**Section 3.1.3**
Does $T$ depend on $s$? It should not as I understand it (should be an automatically adjusting hyperparameter of the learning algorithm), but here it is said that $T$ is an additional prediction head of the policy network, which is conditioned on $s$.

**Section 4**
How does the integration of the safety policy into the hybrid policy compare with training with a shield which ensures the action is safe after it is predicted by the agent, and replaces it otherwise? Because $\pi_\theta$ can be learned and can compensate $\pi_{safe}$ in $\pi_\lambda$ for any $z(s) < 1$, I believe this is actually equivalent to a shield which would let $\pi_theta$ untouched until $z(s) = 1$ and then take over with $\pi_{safe}$.

**Section 5.2**
This is a positive comment: I really enjoyed this section!

**Section 6**
How is the runtime safety shielding (“Simplex”) generated? How is the safety checked, and which safety action is enforced

**Section 6.1**
I do not understand how the safe policy (for Simplex) intervenes in a “non-stationary” way. I also do not understand how this is different from Autosafe, except for the smoothening.

**Section 6.2**
Here, it seems $T$ is not state-based (see question on 3.1.3). Still, I don’t understand the analysis. As $\pi_\theta$ could completely absorb $\pi_safe$ for any $z(s) < 1$, I believe the value of $T$ at convergence should have no impact on the performance. Overall, this complexifies the learning process without a solid explanation for the better performance at convergence observed in the results. Could it be related to a smoother learning process, such as $\pi_{safe}$ guiding the learning at the beginning because it has a relatively good performance compared to random policies?

**Section 6.3**
Given the assumptions over the systems for which Autosafe can be applied to, I think “holding the promise of bridging domain gaps such as sim-to-real and offline to online RL” is a bold claim.


## Typo/clarity comments:

Section 3 - there is section 3.1, 3.1.1, 3.1.2, 3.1.3, but not 3.2. Could it be restructured without the .1. ?

Section 3.1.3 - it could be worth reiterating that $\lambda$ is capped at 1 here, even if z can be higher than 1.

Section 4 - typo at **T**he stochastic action (capital T is not needed)

---

> ### Author Response · Authors · 2025-11-27
> **Response to Weaknesses**
>
> We thank the reviewer for the detailed feedback and valuable questions. We would like to clarify some ambiguities and provide more details in the following responses.
>
> ### W1: The prerequisite of (1) the dynamics are (approximately) known and (2) the dynamics and the constraints can be expressed in a linear manner should be explicitly stated in the introduction and abstract, ...... The paper would benefit from a less shiny, more rigorous packaging.
>
> A: We agree on this point. The current proposed architecture targets the problem where the system model and safety constraints are known a priori and can be linearly approximated. This is motivated by the fact that most of the control applications have the linear approximation of the system model $\dot{s} = As + Ba$ and the linear constraints approximation (over-approximation for the nonlinear case for ensuring safety. Those linear approximations can solve a large set of applications; **nevertheless** they can not solve all Safe RL problems, for instance, the constraints are highly combinatorial or multilayered.
>
> Although we mentioned that the considered constraints are polytopic, the solution of $P$ and $K$ requires knowledge of the system dynamics in Eq. (3), and the solved safe policy is a feedback control law $K$ in Sec. 2.2; these aspects might not be sufficiently clear. We have explicitly incorporated the linear perspective of the model and safe policy in the introduction. Thanks for the suggestions to make the paper more rigorous.
>
> We refer to the safety common sense as (1) recognizing the risk, (2) taking safe actions correspondingly. In the current presented AutoSafe architecture, (1) and (2) are both handled by considering the linear model and linear constraints, as they are the most fundamental and cleanest form with **formal** safety analysis in nowadays practical applications. In less safety-critical applications, we believe that the current requirement for linear design can be relaxed and the safety design can be enhanced with a learning-based risk evaluation module and a learning-based safe policy to improve overall safety assurance.
>
> ### W2: The contribution itself is not clear....the proposed method is very similar to the general framework of (white-box)....where prior knowledge of the world is used to determine if the action is safe and otherwise replace it with a safe action.
>
> A: The general white-box framework outputs a binary decision: safe or unsafe, by checking whether the predicted state violates the safety constraints or not. If the action results in a safety violation, a backup action will be taken. Our architecture works in a similar principle, but offers the following different advantages:
>
> Compared to other typical white-box framework outputs, such as the CBF-based approach and shielding, etc.
>
> 1. A new perspective on safe action backup mechanism via safe action blending, which is complementary to the existing methods, including action replacement, action projection, and action masking [1].
>
> 2. We don't need a model for online safety checking and optimization for generating safe actions. Instead, we directly check the safety evaluation and generate action through simple matrix multiplication. Our method is more computationally efficient and always has a solution for safe action. Online optimization-based methods require more computation and may not always yield a valid solution.
>
> Innovation beyond Standard Simplex Architecture:
>
> 1. Smooth transition from unsafe to safe.
> 2. Guided exploration at the beginning of the training.
> 3. The data is generated from a single policy; however, in simplex, it is effectively as if there are two policies interacting with the environment. This significantly impacts policy learning, particularly for stochastic policies. See more discussion below.
>
> ### W3: The authors refer to “non-stationarities” in the simplex learning,
>
> By “non-stationarities,” we intended to refer to the discontinuities in the transition data caused by the hard switch. These abrupt changes make it more difficult for the neural networks to approximate the Q-function and the policy, even though the underlying MDP itself remains stationary. To avoid confusion, we will replace the term “non-stationarities” with data discontinuities. This issue has also been noted in prior work [2], where hard switching can induce control chatter and erratic system behavior.
>
> To further clarify the role of the “Simplex” mechanism, we have added detailed implementation descriptions for each case study in Appendix D.
>
> [1] Krasowski, Hanna, et al. "Provably Safe Reinforcement Learning: Conceptual Analysis, Survey, and Benchmarking." Transactions on Machine Learning Research (2023).
>
> [2] Hsu, Kai-Chieh, Haimin Hu, and Jaime F. Fisac. "The safety filter: A unified view of safety-critical control in autonomous systems." Annual Review of Control, Robotics, and Autonomous Systems 7 (2023).

---

> ### Author Response · Authors · 2025-11-27
> **Response to Questions  Part I**
>
> ### Introduction:
> Yes, building a "good" shield that is not overconstraining while providing safety guarantees is challenging, as discussed on "perfect filter " in Section 2.3 of [1]. Because we lack a full understanding of the system dynamics and how the agent behaves. Therefore, it is difficult to design a good principle to achieve "perfect". This actually motivates this paper. We only enforce the boundary condition (in the worst case) and allow the agent to learn from interactions to decide when and where to let the safe policy intervene, while still maximizing performance. To construct the safety shield with formal properties (the boundary condition in our case), prior knowledge is necessary.
>
> [1] Hsu, Kai-Chieh, Haimin Hu, and Jaime F. Fisac. "The safety filter: A unified view of safety-critical control in autonomous systems." Annual Review of Control, Robotics, and Autonomous Systems 7 (2023).
>
> ### Section 2.2:
> Yes, $\dot{s}$ is the first-order derivative of $s$, and the equations describe how the system's states evolve. We add the explanation for $\dot{s}$ in this section to make it more understandable.
>
> ### section 3.1:
> We thank the reviewer for their question regarding the action space. In our environment, the action corresponds to an actuator command (e.g., torque or motor effort). As is standard in continuous-control benchmarks (Gymnasium, MuJoCo, PyBullet, Isaac Gym), the action space is typically defined as $\mathcal{A} = [-1,1] * a_{max}$. This interval is compact, convex, and symmetric, which aligns with the actuator model of the system and is necessary for policy optimization. The compact and continuous action space setting is also physically realistic for most of the continuous control tasks in practice.
>
> ### section 3.1.3:
> $T$ is dependent on observations $o$ which includes the state $s$ and other information $i$. The policy network $\pi_{\theta}$ takes $o$ as input, outputs the action vectors and the value of the temperature parameter $T$. The motivation is that the agent decides on each state which action is safest to use that can maximize the performance while being safe.
>
> ### Section 4:
> Our main innovation beyond the shield is the smooth, safe action blending. Although it may seem simple, it offers several desirable properties. See more detailed discussion in the response to W2.
>
> ### Section 5.2
> Thanks for the positive feedback.
>
> ### Section 6
> The simplex leverages the same safety matrix $P$ and safe policy $K$. The standard simplex (shielding) working principle can be summarized as follows:
>
> $a = a_{safe}~\text{if} s^\top P s \ge 1, a = a_{drl}, otherwise$
>
> It means if the system is escaping the safety envelope, the safe policy will be used to safeguard the system.  We found that this procedure often requires manual adjustment to stabilize the system's behaviors. When considering an online policy learning setting, a stable data distribution is preferred for the algorithm to converge. To this end, the system needs to behave more smoothly. Our architecture enables smooth transitions from unsafe to safe via the ERA module, showing faster convergence than the other baselines. We believe this feature could be helpful in an online continuous learning setting, as demonstrated in the real-world experiments. We provide a detailed explanation for Simplex in Appendix D.

---

> ### Author Response · Authors · 2025-11-27
> **Response to Questions Part II**
>
> ### Section 6.1
> The MDP is still stationary. The stationarity is meant for the distribution of the collected transitions. As discussed in the response to W3.
>
> ### Section 6.2
> $\pi_{\theta}$ could completely absorb $\pi_{safe}$ but it is not always necessary. If $\pi_{\theta}$ completely absorbs the $\pi_{safe}$, which means that the $\pi_{safe}$ is completely destructive with respect to the performance maximization.
>
> In fact, $\pi_{safe}$ is a suboptimal policy that can still be very useful, especially at the beginning of training. Consider a policy training from scratch, the early exploration would be very stochastic and violate the safety constraints. With the guidance of the safe policy, the agent is exploring the environment more safely and meaningfully. The design of the exponential ramping activation grants the flexibity to $\pi_{\theta}$, which allows to weighting down the contribution of $\pi_{safe}$ to avoid the bias introduced by the suboptimal behaviour of hte $\pi_{safe}$. The benefits of using a model-based policy prior can also be noticed in the literature [1] [2]. However, in [1][2], the policy prior is mainly considered for improving sampling efficiency, where the safety assurance is not discussed.
>
> [1] Cheng, Richard, et al. "Control regularization for reduced variance reinforcement learning." International Conference on Machine Learning. PMLR, 2019.
>
> [2] Johannink, Tobias, et al. "Residual reinforcement learning for robot control." 2019 international conference on robotics and automation (ICRA). IEEE, 2019.
>
> ### Section 6.3
> Thanks for pointing out the concern. We have added **for continuous robot control tasks** to limit the scope. Within this scope, the primary challenges of sim2real and offline2online problems are the domain gaps resulting from dynamics shifts, out-of-support data (i.e., unseen inputs), and task misalignment. For out-of-support inputs, our method is not restrictive, as it is data independent. For the unknown dynamics shift, we would need to consider the model mismatch in the model linearization (see the response to W1 from reviewer mswY) to design a robust and safe policy. When deployed to a target domain, our architecture enables agents to interact safely with the environment, allowing the robot to gradually expand its acting space and improve itself to accomplish new tasks.
>
> ### Typo/Clarity Comments
> We have revised all the raised suggestions in the revised version. Thanks for the detailed feedback.

---

### Official Review · Reviewer_mswY · 2025-10-31

**Soundness:** 3
**Presentation:** 3
**Contribution:** 3
**Rating:** 4
**Confidence:** 2

**Summary:**

This paper proposes a novel safe reinforcement learning (DRL) architecture named AutoSafe, which aims to embed "safety common sense" directly into the neural policy to address DRL guarantee issues in safety-sensitive applications. The core idea of AutoSafe is to fuse a traditional, high-performance DRL policy ($a_{\theta}$) with a model-based, verifiable safe policy ($a_{safe}$) through a smooth and adaptive mechanism.

**Strengths:**

- Novel Internal Fusion Architecture: Unlike traditional methods relying on external "safety shields" for hard switching (e.g., Simplex) , AutoSafe integrates the safety evaluation and correction modules as internal components of the policy network itself. This design makes the entire policy end-to-end differentiable and easy to integrate into existing Actor-Critic algorithms (like SAC).
- Real-World Applicability Validated: The paper does not stop at simulation. It successfully deploys AutoSafe on an embedded device (Raspberry Pi) for a physical Cartpole system. This experiment demonstrates the architecture's ability to conduct safe, continual learning  under real-world physical noise and delays, greatly enhancing the method's credibility and practical value.

**Weaknesses:**

- Strong Dependence on Linear Models: The entire safety foundation of AutoSafe is built upon the pre-calculated, model-based SCM ($K$ matrix) and SEM ($P$ matrix. As shown in Appendix A, these matrices are obtained by solving an LMI, which depends on a known, linear system dynamics model ($\dot{s} = As + Ba$). This prerequisite may not be met for many of the highly non-linear, difficult-to-model complex problems that DRL aims to solve.

**Questions:**

- Dependence on State Vector $s_t$: AutoSafe's safety core (SEM and SCM) relies heavily on a precise, complete, and low-dimensional state vector $s_t$ as input. If the agent can only learn from high-dimensional observations (e.g., pixels), how would this architecture obtain the $s_t$ needed to calculate the risk $z_t$? Would it require an additional (and possibly imperfect) state estimator? How would errors from this estimator affect AutoSafe's safety guarantees?
- Performance under $a_{safe}$ Goal Conflict: The paper analyzes $T$'s adaptiveness when $a_{safe}$ is "suboptimal" in performance. But what happens if the goal of $a_{safe}$ (e.g., stabilize at the equilibrium $s^*=\{0\}$) is in fundamental conflict with the DRL reward goal (e.g., reach a target $\hat{s}=\{0.1\}$)? Could this internal conflict lead to high-frequency oscillations between the two objectives, or cause the DRL policy to fail to learn?

---

> ### Author Response · Authors · 2025-11-27
> **Response to Weaknesses**
>
> We thank your positive feedback on the novelty and real-world applicability. We also thank your constructive suggestions and questions. We address them below:
>
> ### W: Strong Dependence on Linear Models. The prerequisite may not be met for many highly complex, difficult-to-model problems that DRL aims to solve.
>
> A: We agree that finding a model linearization approximation for very complex systems is non-trivial. In fact, the complexity of the problem is coming from two perspectives: dynamic complexity and task complexity.
>
> To address the dynamic complexity, i.e, to make the model linearization approximation effective for the complex non-linear system. Two typical procedures can be used to address this concern:
> 1. Stability preserving linearization: When linear approximates the nonlinear model, we consider a worst-case model mismatch in the linear approximation. Such that the stability properties of the linear model also hold for the non-linear true system. As demonstrated in our experiments and other related works [1], the derived safe policy can generalize to nonlinear, complex cases, such as a complex quadruped robot.
>
> 2. Model relinearization: When the system operates in different modes within different state spaces (underlying dynamics changes), we can create a bank of approximated models (or online MPC-like [2])  to cover the operating space, and according to the state, we switch to the corresponding linear model with the corresponding safe policy design to safeguard the system. Furthermore, our architecture employs a safe policy to regulate the system's operation within the safety envelope, thereby ensuring the model error remains small.
>
> To address task complexity, we explicitly separate safety objectives from task-oriented objectives and rely on a linear model to design the safety matrix $P$ and policy $K$, focusing solely on meeting the safety requirements, where the DRL is deployed to achieve high-performance task requirements. From the perspective of task complexity, the linear model is not a bottleneck and does not affect the task-related objective.
>
> [1] Cao, Hongpeng, et al. "Physics-Regulated Deep Reinforcement Learning: Invariant Embeddings." The Twelfth International Conference on Learning Representations.
>
> [2] Grandia, Ruben, et al. "Multi-layered safety for legged robots via control barrier functions and model predictive control." 2021 IEEE International Conference on Robotics and Automation (ICRA). IEEE, 2021.

---

> ### Author Response · Authors · 2025-11-27
> **Response to Questions**
>
> ### Q1: Dependence on State Vector
> A: The State vector is explicitly needed in the proposed architecture due to the principled safety design, including safety evaluation $z = s^\top P s$ and safe correction $a_{safe} = Ks$.
>
> From a practical point of view, most of the robot is equipped with an IMU sensor or positional encoders, etc that can directly obtain the state vector in the body frame. Global state estimation (odometry estimation) is also a widely studied direction that can provide state estimation directly from visual information or other modalities [1].
>
> In fact, it is very hard to provide safety assurance for a direct vision-based control architecture. Almost all safe architectures with formal safety assurance require access to the state vector [2].
>
> [1] Scaramuzza, Davide, and Friedrich Fraundorfer. "Visual odometry [tutorial]." IEEE robotics & automation magazine 18.4 (2011): 80-92.
>
> [2] Hsu, Kai-Chieh, Haimin Hu, and Jaime F. Fisac. "The safety filter: A unified view of safety-critical control in autonomous systems." Annual Review of Control, Robotics, and Autonomous Systems 7 (2023).
>
> ### Q2: Performance under $a_{safe}$ Goal Conflict:
> A: This is an open question in current safety filter literature and a less discussed problem in hybrid controller literature, for instance, residual architecture [1], mixed policy architecture [2]. Our architecture aims to contribute to this problem by using a smooth action mixing mechanism, called Exponential Ramp Activation (ERA). ERA includes a learnable parameter $T$, which determines the sharpness of an exponential function. We found that the noticeable conflict only appears when the DRL goal is very close to the boundary of the safety envelope; it inevitably triggers the safe controller, making learning more challenging (see the theoretical analysis in Section 5.2 and the experimental results in Figure 5 and the revised Appendix E.6).
>
> Moreover,  the results presented in Figure 5 justify your question. In this experiment, we intentionally deviate from the control goal of the model-based design. The temperature $T$ is converged to different values, yielding different shapes of the exponential curves. As a result, the agent learns to converge to a larger $T$ when DRL goals deviate more from the model-based goal. The larger $T$ means that the agent places more weight on the action from the DRL output and less weight on the model-based action (as it is suboptimal in that case). Although the goals are different, the autosafe agent is trained to maximize the DRL objective, where the model-based safe policy is only used for the safety objective via ERA through enforced boundary conditions. Furthermore, as shown in E.6, the performance of RL degrades when we put the goal of DRL close to the safety envelope's boundary
>
> [1] Johannink, Tobias, et al. "Residual reinforcement learning for robot control." 2019 international conference on robotics and automation (ICRA). IEEE, 2019.
>
> [2] Tian, Haozhe, et al. "Reinforcement learning with adaptive regularization for safe control of critical systems." Advances in Neural Information Processing Systems 37 (2024): 2528-2557.

---

### Author Response · Authors · 2025-11-28
**Global Response**

Dear Reviewers,

Thank you very much for your valuable feedback and constructive questions. In our earlier responses, we have addressed the raised questions and provided more technical details, supported by additional experiments. We have also incorporated the helpful suggestions into the revised version of the paper.

We would like to invite you to review the response details and the updated paper. We hope that our responses and the revised manuscript address your concerns. We look forward to any further discussion with you.

Best regards,

Authors

---

### Meta-Review · Area_Chair_2aG8 · 2025-12-30

**Summary:**

The reviewers agree that the contribution is somewhat novel and significant for safe RL.

The reviewers note that the paper makes strong assumptions about the dynamics of the underlying system (linear dynamics and linear constraints) and about the system itself. Furthermore, the paper is not so clear about that, even after the rebuttal. Lastly, they raised concerns about the number of baselines.

The paper could use a more straightforward presentation of the setting, make an explicit comparison with related work, and strengthen the empirical evaluation by adding more baselines.

**Reviewer Concerns:**

- `mswY` is mainly concerned about the strong dependency on linear models. Although the authors discuss model linearization in the rebuttal, this is insufficient to address such a concern; such approaches have limitations and would not mitigate these dependencies in a highly nonlinear, difficult-to-model problem.
- `otqo` found that the assumptions could be stated more explicitly and the contribution unclear. In general, the rebuttal was somewhat convoluted. Although the rebuttal states that the assumptions were clarified in the introduction, the text does not make this as explicit as the reviewer suggests. The rebuttal only partially addresses the reviewer's concerns about the method's novelty, as some of the claims lack sufficient support. The question regarding the differences from the simplex architecture was clarified well.
- `pSva` (i) questioned the clarity of the problem setting, (ii) mentioned that the empirical evaluation could use other baselines, and (iii) noted that the contribution of the approach was not clear. The rebuttal was sufficient to address (i). During the rebuttal, new experiments were added to the appendix of the paper, which only partially addressed (ii); in particular, it is unclear why such results were not included in the main document. Lastly, (iii) the novelty of the approach still remains incremental, and the point of the concerns of the reviewer remains open after the rebuttal.

**Reviewer Scores:**

- `mswY`: 4 -> 4
- `otqo`: 2 -> 4
- `pSva`: 2 -> 4

---

### Decision · Program_Chairs · 2026-01-26

Reject